# The *Ex Situ* Conservation and Potential Usage of Crop Wild Relatives in Poland on the Example of Grasses

Denise F. Dostatny [1,*], Grzegorz Żurek [2], Adam Kapler [3] and Wiesław Podyma [1]

1 National Centre for Plant Genetic Resources, Plant Breeding and Acclimatization Institute—NRI, Radzików, 05-870 Błonie, Poland; w.podyma@ihar.edu.pl
2 Department of Grasses, Legumes and Energy Plants, Plant Breeding and Acclimatization Institute—NRI, Radzików, 05-870 Błonie, Poland; g.zurek@ihar.edu.pl
3 Polish Academy of Sciences, Botanical Garden in Powsin, Prawdziwka 2, 02-973 Warsaw, Poland; a.kapler@obpan.pl
* Correspondence: d.dostatny@ihar.edu.pl

**Abstract:** The *Poaceae* is the second most abundant family among crop wild relatives in Poland, representing 147 taxa. From these species, 135 are native taxa, and 11 are archeophytes. In addition, one taxon is now considered to be extinct. Among the 147 taxa, 8 are endemic species. Central Europe, including Poland, does not have many endemic species. Only a few dozen endemic species have been identified in this paper, mainly in the Carpathians and the adjacent uplands, e.g., the Polish Jura in southern Poland. The most numerous genera among the 32 present in the crop wild relatives (CWR) of *Poaceae* family are: The genus *Festuca* (33 species), *Poa* (19), and *Bromus* (11). In turn, ten genera are represented by only one species per genus. A good representative of groups of grasses occur in xerothermic grasslands, and other smaller groups can be found in forests, mountains, or dunes. CWR species from the *Poaceae* family have the potential for different uses in terms of the ecosystem services benefit. They can impart for humans, animals, and the environment, including fodder, edibles, biomass grasses (fuels and raw material), and amenity grasses, and are used for ecological purposes. In the central Polish gene bank in Radzików (NCPGR), all accessions are represented by approximately 30% of grasses germplasm, 10% of which are CWR grasses. In the case of CWR grasses, 56% are stored in the NCPGR gene bank, and approximately 80% in botanical gardens, but frequently in a single accessions. Together, crop gene banks and botanical gardens can maintain a large range of ex situ collections useful for the preservation, breeding, and research of crop wild relatives along with the necessary information for plant breeders.

**Keywords:** *Poaceae* family; potential usage; conservation; gene bank; botanical gardens

## 1. Introduction

Crop wild relatives (CWR) include some crop ancestors as well other species more or less closely related to crops. A wider spectrum of diversity can be found in the genomes of wild plant species. Russian biologist and geneticist, Mikołaj Iwanowicz Wawiłow, underlined the importance of maintaining genetical diversity of crop plants in cultivation, and making use of the crop wild relatives to enhance this diversity through breeding [1]. The views were formalized by Harlan and de Wet [2], who classified each crop and its related species by gene pools according to reproductive barriers. The application of the concept in practice is only possible when knowledge is available of the structure of genetic diversification among the species in question and their ability to crossbreed. Therefore, it has been proposed to use the existing taxonomic hierarchy to define "crop wild relatives" as "a wild plant taxon that has an indirect use derived from its relatively close genetic relationship to a crop plant" [3]. According to Maxted et al. [3], this relationship corresponds to the following taxonomic hierarchy (Taxa Groups): TG1a—crop taxon; TG1b—same species as crop; TG2—same series or section as crop; TG3—same subgenus

as crop; TG4—same genus as crop; and TG5—same tribe, but different genus of crop. This strategy has been applied within the European CWR project—PGR (Plant Genetic Resources) Forum. It was the first project related to the knowledge of European CWR, continued through activities undertaken within the context of projects initiated by members of the ECPGR (European Cooperative Programme for Plant Genetic Resources) In Situ and On-farm Conservation Network [4].

Monoculture and the resulting impoverishment among cultivated species, as well as the necessity to adjust crops to better withstand climate changes, has brought about commencement of work in many countries on preservation and the assessment of native vascular flora with regard to the possibilities of using its representatives in the breeding of new species of crops or diversifying those which have already been cultivated. The first stage of the work included drawing up lists of crop wild relatives. The CWR checklists, as well their conservation strategy, have already been completed in several countries. These include: Great Britain [5], Finland [6], Italy [7], Portugal [8], the Czech Republic [9], Poland [10], Spain [11], Mexico [12], the Republic of Southern Africa [13], Malawi [14], and China [15], among others.

Climate change and species invasions are probably the most detrimental factors affecting future global food security [16–18]. To reduce their negative impacts, humankind will require crops to be more genetically diverse [19]. Necessary genetic diversity has been lost during domestication and stayed available only within crop wild relatives gene pools. In the new era of gene editing, advantageous traits can be transferred directly to the cultivated plant from any wild taxa, but the closely related congeners remain the optimal source of beneficial traits. Preservation of CWRs genetic resources ought to reside within the nation where they occur at natural stands [20]. Therefore, prioritization for the conservation of crop wild relatives based on the Polish CWRs checklist is crucial for the Polish and global food security [10]. CWRs remain inestimable sources or crop improvement in tackling both abiotic and biotic stresses. As we face pronounced climatic changes with erratic and extreme weather patterns, the invasion of new pests, weeds, and pathogen species, this is becoming increasingly important [21–23].

Recognizing a crop wild relative as a potentially useful source of new variation for breeding depends on its gene pool availability. Taking into account the techniques of "naked" DNA manipulation known as genetic engineering, presently there are no limitations in the possibilities of gene transfer, and each breeder has all the genes from all of the species, and not only plant species, at their disposal [3,24]. However, if only the classic breeding methods that utilize sexual reproduction are to be used, the possibility to cross the barriers of interspecific isolation becomes important. In such a case, it is important to know whether the list of species useful for breeding includes ones with which hybrids have been produced, if they produce fertile hybrids that allow them to build a permanent alien chromosome, or at least its fragment, into the genome. It is important to bear in mind that in natural science there is no possibility in precisely stating the impassability of the interspecific barrier, both in relation to the impossibility to cross-breed and the sterility of hybrids. The statement that species do not cross usually means only that up to now, no such genotypes and/or no such conditions of parent form treatment have been found that would enable a hybrid to be produced. To a large extent, the same is the case with hybrid sterility, which also depends on genotype and environment. The probability of success depends directly on the number of genetically varied components used for crossbreeding, and for sterility—also on the extent of multiplication of the interspecific F1 hybrid.

Historically, grassland and some types of forage crops played a major role in the agricultural development in most parts of Europe [25,26] compared to nowadays. In the past, many more grass species achieved some economic importance. For example, in Poland, floating sweet-grass *Glyceria fluitans* has been gathered from the wild populations as food and medicine as early as the 1380s. In the 2nd half of the 18th century, attempts were made to domesticate it in Hungary, the Czech Republic, and Germany [27]. Nowadays, this

species could potentially be used in making processed, gluten-free, delicacy food products, as well as enormously productive fodder [28].

This paper presents all CWR taxa belonging to the *Poaceae* family from the Crop Wild Relatives Polish checklist—"Crop wild relatives occurring in Poland. Checklist, resources and threats" [10], which constitutes the second most abundant CWR family in Poland. Their habitats are given, and those most endangered are highlighted. The functional features of individual species, their economic potential, and their suitability for the creation of new varieties, are presented. The *ex situ* conservation status of these taxa is analyzed, both in Polish gene bank collections (National Center for Plant Genetic Resources and Kostrzyca Forest Gene Bank) and in seed bank collections (these mainly include collections of seeds of endangered species kept in botanical gardens—regional) located in different institutions, as well as in field collections of botanical gardens. The main goal of this study is to illustrate the diversity of CWR grasses in Poland, their preferred habitats and potential uses, as well as gaps in the *ex situ* collections. The article also aims to inform breeders about the potential of this group of plants and to encourage them to use genetic grass material in breeding programs.

## 2. Materials and Methods

The list of crop wild relatives taxa belonging to the *Poaceae* family was taken from the "Crop wild relatives occurring in Poland. Checklist, resources and threats" [10]. The study entitled "Flowering plants and pteridophytes of Poland—a checklist" by Mirek et al. [29] constituted the starting point for drawing up the list. Because of differences in the contemporary taxonomic presentation of some plants, the whole list was verified with "The Plant List" [30] database. However, in the case of low rank taxa, the "Flowering plants and pteridophytes of Poland—a checklist" [29] was considered.

In the study entitled "Crop wild relatives occurring in Poland, checklist, resources and threats" [10], the authors decided to omit the regional endemics and other taxa with rank lower than species (which are not included in the international databases) in order to avoid elongation of the checklist. However, this paper also includes a subspecies for those listed in order to show the importance between such species as endemics. In the same study, taxa belonging to the "TG1a" were excluded from the checklist, and therefore in this article there are no taxa of cultivated crops, e.g., cereals. Crop species have been identified as cultivated plants if they are included in "The Polish National List" [31] or identified as a crop in the "Flowering plants and pteridophytes of Poland—a checklist" [29] and also in other sources, such as the materials presented in the Polish Nurserymen Association website (inventories of ornamental plants grown in Poland) [32]. Crop plant species have been included on the list only if it has been confirmed that they also occur in the wild in the territory of Poland (TG1b, according to Maxted [3]). The floristic status (native/archeophyte) in Polish Flora was verified (Figure 1) according to "Flowering plants and pteridophytes of Poland—a checklist" [29] and "Rośliny obcego pochodzenia w Polsce ze szczególnym uwzględnieniem gatunków inwazyjnych" (=Alien plants in Poland with particular reference to invasive species) [33].

Taxa were grouped and assigned to 13 habitats according to Matuszkiewicz [34].

Ellenberg's indicator values were applied to the taxa of the CWR grasses in three gradients: Light (L), soil moisture (M), and reaction—soil acidity/pH (R), based on Zarzycki et al. [35].

Species were verified by category of threats on the basis of the following studies:

✓　European Red List of Vascular Plants [36] and European Red List of Medicinal Plants [37];

✓　Polish Red List of Pteridophytes and Flowering Plants [38] and Polish Red Data Book of Plants [39];

✓　recognizing (direct use) and potential usage of species in Poland were classified according to the following categories:

✓　Fodder (FO)—plants which could be eaten by wild animals of farm livestock;

✓     edible (ED)—plants used by humans as an additive in other dietary elements;

✓     the biomass grasses (BG)—taxa can be used as sources of renewable energy (direct combustion, biogas, conversion to ethanol, etc.) or for different industrial purposes, i.e., pulp for paper production, as a component of construction or insulation elements;

✓     amenity grasses (AG)—the possible (or actual) application is for home, sport, or other natural lawns, green surfaces, for landscape areas, road banks, etc. Grasses can be used as a specimen plant in perennial flower beds or in large groupings and mass plantings in gardens, parks, and recreation areas. Whole plants or panicles can be used in floristry in dry flower arrangements;

✓     ecology (EC)—i.e., plants suitable to be grown in a wide range of soil conditions including waste and polluted areas, for land reclamation and/or habitat restoration and soil stabilization. Species for increasing biodiversity on arable lands, i.e., green areas as grassy strips on the fields (field margins).

The species included in the list were also verified with the following criteria:

✓     Presence in long-term storage in the National Centre for Plant Genetic Resources in Poland and in the Kostrzyca Forest Gene Bank, as well as in other regional seed banks;

✓     presence in botanical gardens and arboreta as listed in three studies: "Index *Plantarum* Polskich Kolekcji Dendrologicznych" (=Index *Plantarum* of the Polish dendrological collections) [40], "Index *Plantarum* of Outdoors Cultivated Herbaceous Plants in Poland" [41], and "Kolekcje Roślin Chronionych i Zagrożonych oraz Gatunków Objętych Konwencją Berneńską w Polskich Ogrodach Botanicznych" (=Collections of protected law and endangered plants and of species protected by the Bern Convention in the Polish Botanical Gardens) [42].

All this data above have been shown in Appendix A.

A taxa map of the CWR *Poaceae* family, endemic species, and endangered species was drawn. On a $10 \times 10$ km grid, all taxa were marked. In each of the presented maps, the number of species in a group were reflected by dots of various sizes. The size depends on the number of species whose habitats occurred in the individual squares. The maximum number (MAX) is the number of taxa occurring in the most representative square (with a higher number of species). They correspond to 4 thresholds (T). They are the following: CWR *Poaceae* family map (Figure 2): below 20 taxa (T1), from 21 to 39 taxa (T2), from 40 to 60 taxa (T3), above 61 taxa (T4), MAX 87; endemic species map (Figure 3): 1 taxon (T1), 2 taxa (T2), 3 taxa (T3), MAX 3; endangered species (Figure 4): Below 4 taxa (T1), from 5 to 8 taxa (T2), from 9 to 12 taxa (T3), above 13 taxa (T4), MAX 18. The maps were drawn based on Atlas rozmieszczenia roślin naczyniowych w Polsce (=Distribution Atlas of Vascular Plants in Poland) [43].

The statistical correlations between species and their habitats were calculated, and their numbers were given according to Zarzycki [35]. Ellenberg indicator values (EIVs) are widely used in vegetation sciences, allowing for the assessment of environmental variables such as light availability (L), soil moisture (M), and soil reaction (R), using in this study, without direct, physical measurements. They've been derived from numerous, detailed vegetation surveys to reflect habitat conditions restricting the occurrence of species or entire plant communities at the studied site. EIVs are based chiefly on field observations and expert knowledge. The Ellenberg indicator system could be a very valuable tool for habitat calibration, provided the appropriate parameters are considered [44–48]. The hierarchical cluster analysis was performed in order to identify groups of taxa among grasses which occur in similar habitats generated by both the Euclidean distance and UPGMA (Unweighted Pair Group Method with Arithmetic mean). A PCA (Principal Components Analysis) was carried out in this study. Based on numerical and graphical results, it was possible to make conclusions, which would not have been possible from a simple observation of the largest data table.

## 3. Results and Discussion

The Polish checklist of crop wild relatives includes representatives of 98 families, of which *Poaceae* is one of the most represented, with 147 taxa [10]. From these species, 135 are native taxa, and 11 are archeophytes (Appendix A), which emerged in the territory of Poland between the Neolithic period and the end of the 15th century (Figure 1). In addition, one taxon is considered to be extinct (in Poland), which was included because of its possible induction into the restoration programs. Among the 147 taxa, 8 are endemic species (Appendix A). The identification of 'areas of endemism' is important for the development and implementation of conservation strategies [49–51]. Central Europe, including Poland, does not have many endemic species. Only a few separate endemic species have been identified in this paper, mainly in the Carpathians and the adjacent uplands, e.g., the Polish Jura in southern Poland, and mostly younger Pleistocene and Holocene neoendemics [52–56].

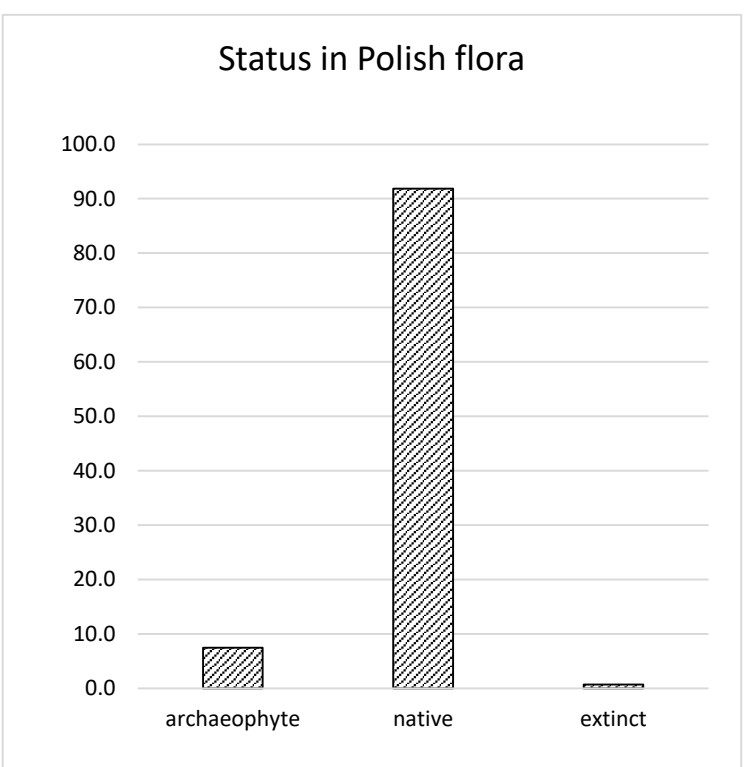

**Figure 1.** Occurrence status of taxa in Polish flora (%).

The most numerous genera among the 32 present in the CWR of *Poaceae* family are: *Festuca* (33 species), *Poa* (19), and *Bromus* (11). In turn, 10 genera are represented by only one species per genus (Table 1).

The taxa of the genera above occur throughout the entire country, but they are not uniformly distributed (Figure 2). Due to high ecological plasticity and an ability to cope with diverse stress conditions and easy transfer of genes from generation to generation, grasses could be found all through the country. A higher frequency of grass species could be noted along the Vistula river and in the Świętokrzyskie region. On the other hand, lower frequencies were located in the dense forest regions, where only a relatively low number of species could exist. Certain taxa are common in Poland and occur in numerous habitats, such as *Agrostis capillaris* and *Festuca rubra*. In contrast, other taxa are rare and occur exclusively in one type of habitat. For example: *Ammophila arenaria* is a typical dune species, *Festuca salina* is restricted to coastal salt marshes, *Festuca carpatica* to limestone rocks, and *Poa nobilis* to granite bedrock.

**Table 1.** Summary of the genera among grasses.

| Genus | No. of Species per Genus |
|---|---|
| *Festuca* | 33 |
| *Poa* | 19 |
| *Bromus* | 11 |
| *Agrostis, Calamagrostis, Glyceria* | 7 |
| *Melica, Phleum, Stipa* | 5 |
| *Alopecurus, Koeleria, Trisetum* | 4 |
| *Agropyron, Deschampsia, Lolium* | 3 |
| *Anthoxanthum, Brachypodium, Dactylis, Digitaria, Elymus, Hierochloë, Holcus* | 2 |
| *Ammophila, Arrhenatherum, Avena, Briza, Corynephorus, Leersia, Milium, Molinia Phalaris, Phragmites* | 1 |

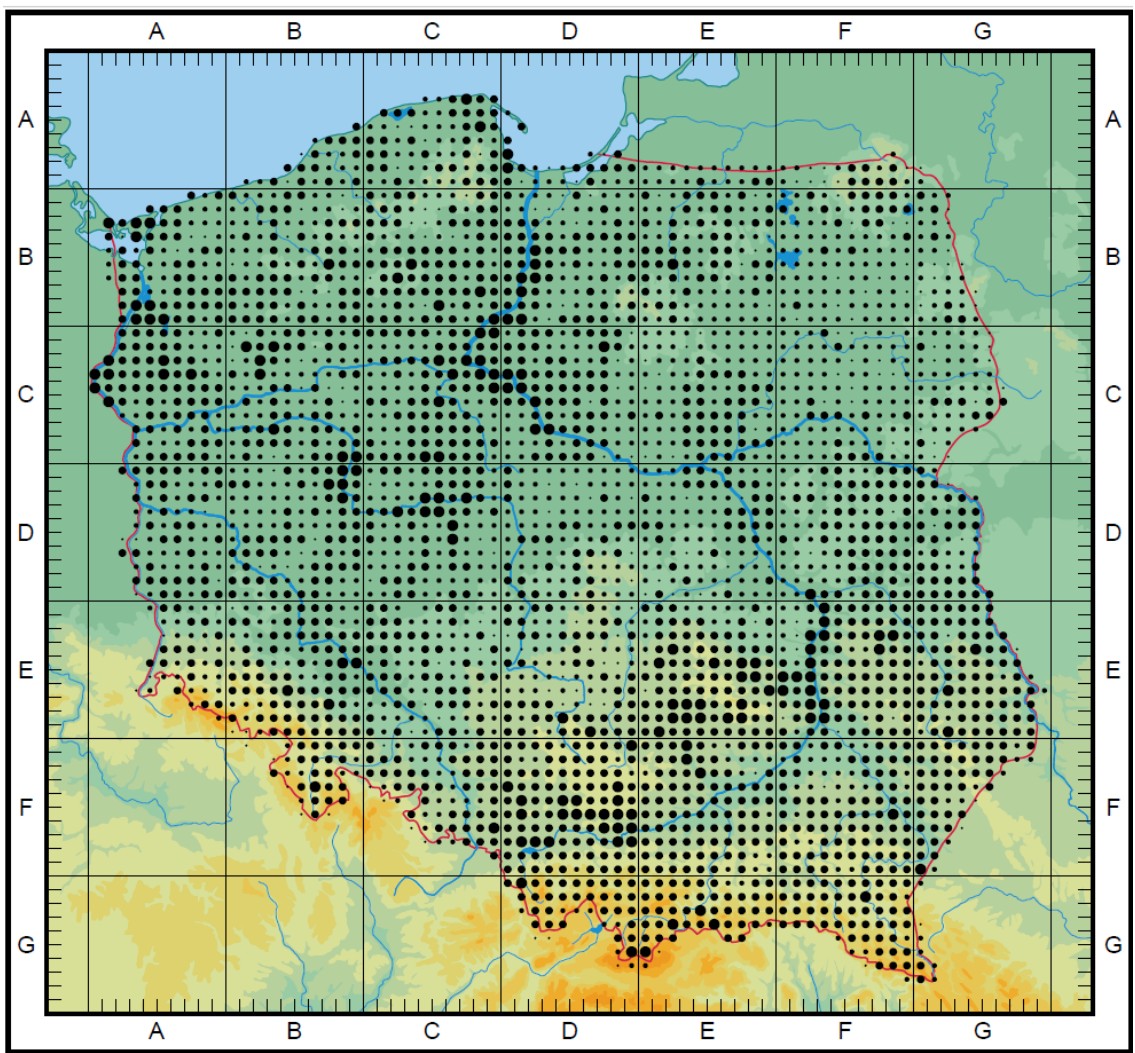

**Figure 2.** The distribution of crop wild relatives of the *Poaceae* family in Poland (map).

In Poland 8 endemic crop wild relatives "regional" taxa occur (often they are not included in the international databases). The distribution of endemic grass species depends on a specific evolution history and could be different in the case of each species. Interesting to note is the fact that the distribution of endemic grass species is connected to the Vistula river near which the vast majority of localities could be found. The following species are

stenochoric: *Festuca carpathica* F. Dietr., *Festuca macutrensis* Zapał., *Festuca tatrae* (Czakó) Degen, *Melica transsilvanica* Schur, *Poa granitica* Braun-Blanq., *Poa nobilis* Skalińska, *Stipa joannis* Čelak. s. s., and *Festuca polesica* Zapał. The last one is a steppe species with an island location, but in this article, it was considered endemic. The aforementioned fescues are alpine species, while *F. macutrensis*, *M. transsilvanica*, and *S. joannis* are a xerothermophilous species. They are restricted to Central Europe and related to the Eurasian steppe species occurring further east. *F. carpatica* remains restricted to alkaline bedrock in the Carpathians. *F. tatrae* is a narrow endemic of the Tatra Mts. *P. granitica* and *P. nobilis* occur only in granite bedrock, *P. granitica* only at Babia Góra, while *P. nobilis* only in the Tatra Mts. [54,55]. There are more endemic taxa with lower or controversial taxonomic rank, usually treated as subspecies or varieties among Polish CWR grasses. The distribution, habitat preferences, and genetic diversity of some of them are fairly well known. These are, e.g., *Festuca amethystina* ssp. *Ritschlii* Hack-Lemke ex Markgr.-Dann., *Stipa pennata* ssp. *Ceynowae* Klichowska, Nobis [57–61] (Figure 3).

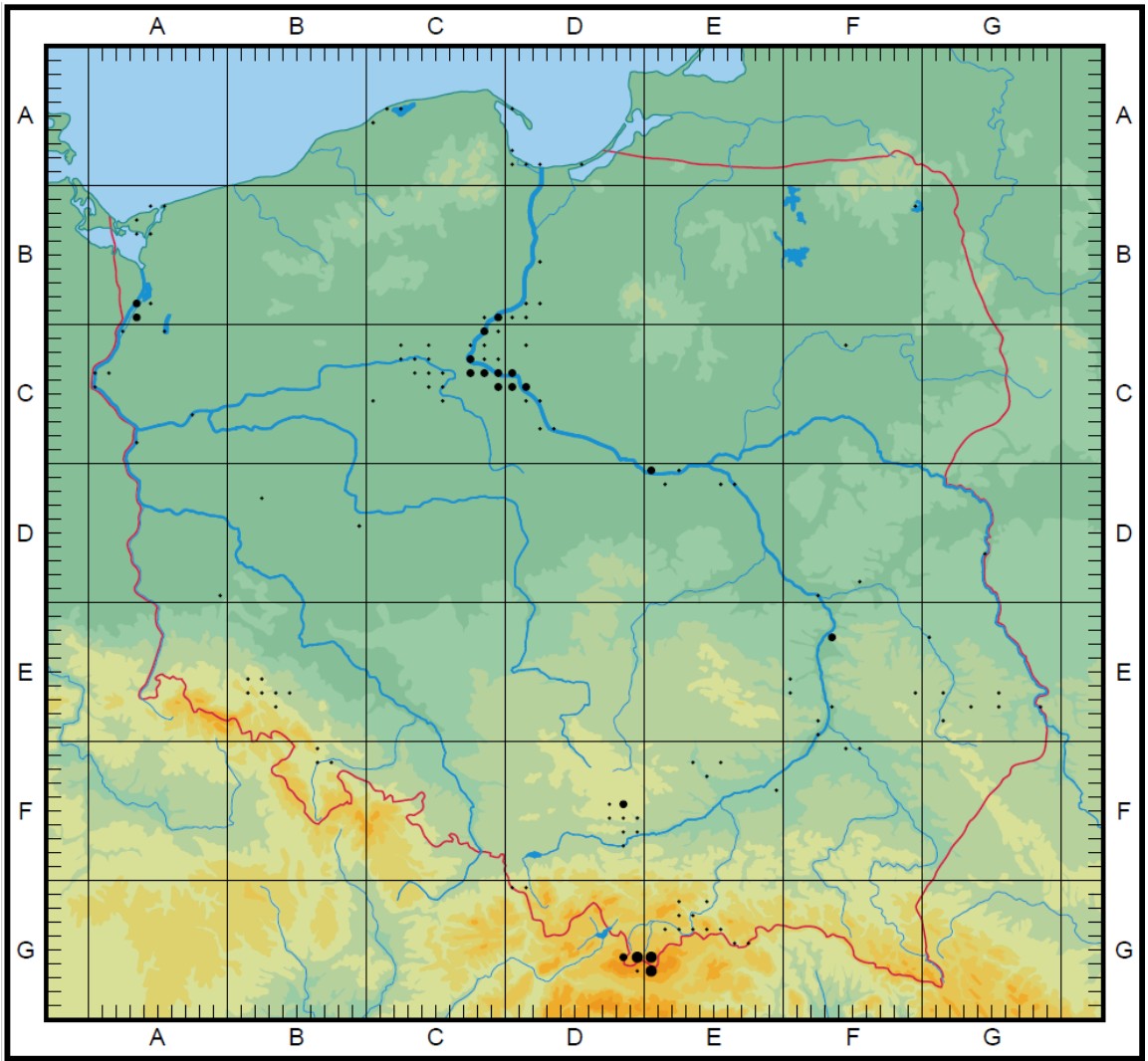

**Figure 3.** Endemic species of crop wild relatives of the *Poaceae* family in Poland (map).

Rare and threatened grass species from the Polish and European Red Data List and Book [36–39] are to be found chiefly in large river valleys (Figure 4). Endangered and rare species occurring in situ represent 29% of all crop wild relative grasses.

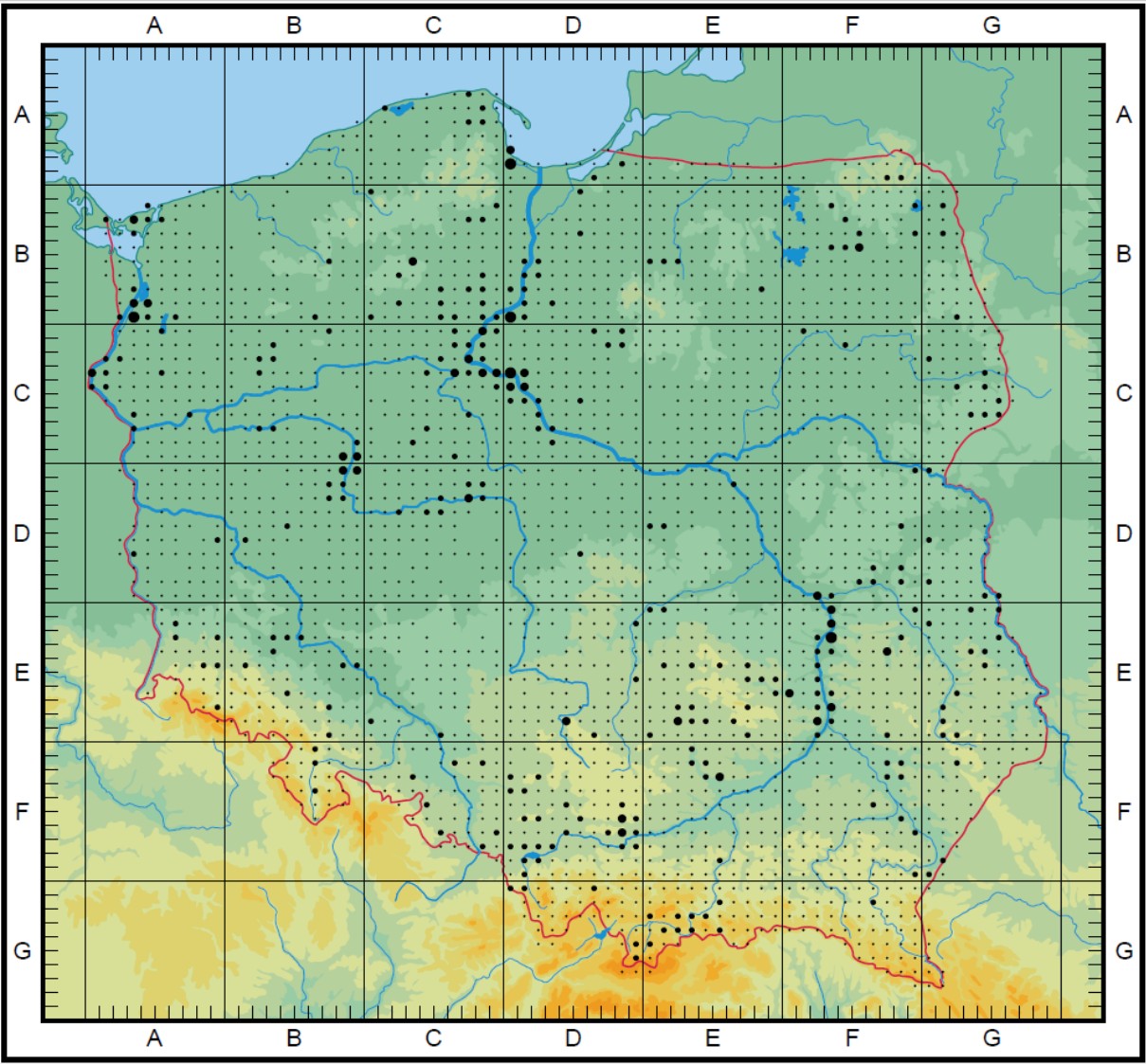

**Figure 4.** Endangered species of crop wild relatives (CWRs) of the *Poaceae* family occurring in Poland and Europe (map).

The results of the hierarchical clustering shows that the first group of species with *Stipa* genus includes species typical for the Eurasian steppe biome and Central European xerothermic grasslands (Figure 5). The second branch of the dendrogram consists of Central and Western European geographic elements. The same branch can be further divided into smaller subgroups of forest, mountain and dune (coastal) floristic elements. Forest and mountain grass CWR species can be further divided into smaller environmental units, restricted to deciduous versus coniferous forests or acidic rocks (granite and similar) versus alkaline rocks (limestone, dolomite) and general-alpine species (indifferent to rock substrate).

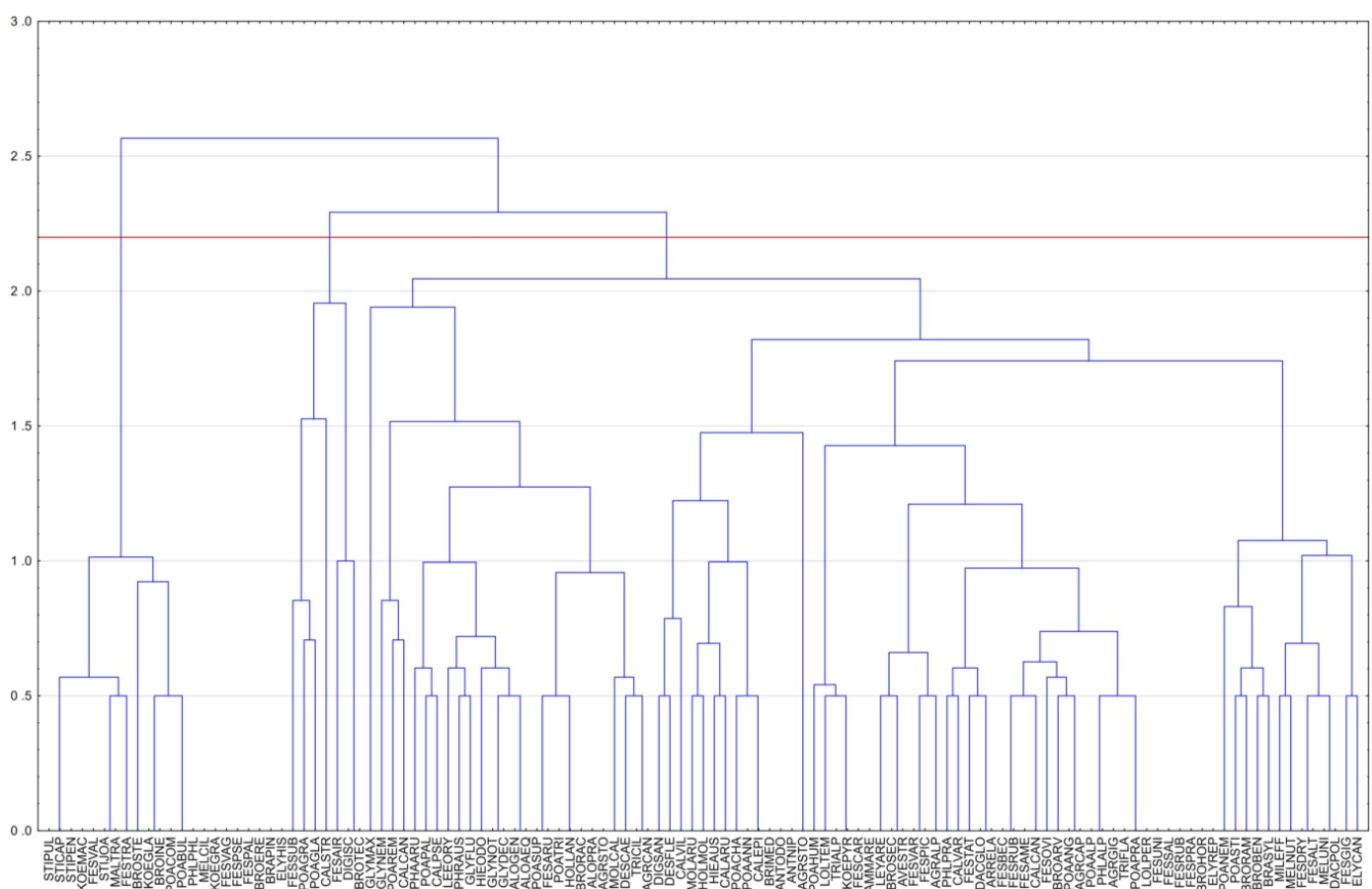

**Figure 5.** Different groups of CWR grass species occurring in Poland according to the Ellenberg indicator values.

Such a division of Polish CWR grass species corresponds to most phytogeographic divisions from the 19th and 20th centuries, dividing Europe, including the lands of Poland and its neighboring countries, into Provinces: Pontic-Pannonian (Eurasian steppes and the Hungarian Puszta); Central European—Mountain (Carpathians and Sudetes in the Pl); and Central European—Uplands and Lowlands (most of the Polish territories, except for the higher mountain ranges). A narrow coastal belt can be distinguished in the latter Province hosting a specific flora of saltpans, dunes, cliffs, and Baltic wet heaths [62,63]. It corresponds roughly to the contemporary official division of Natura 2000 into continental (CON), alpine (ALP) and pannonian (PAN) areas—EU Directive 1992 [64].

Principal component analysis showed that the first two main components accounted more than 80% of the variability observed in the examined group of CWR grasses. The first component, which accounted for 52.6% of the variability, built "light value" (r = 0.79) and "soil moisture value" (r = −0.77). The second component, which is responsible for 27.6% of the variability, was built by "soil acidity value" (r = −80) (Table 2). In relation to the horizontal axis X—PC1, the more to the right the taxon is, the higher the "light value" and the lower the "soil moisture value". In the right axis, we can find typical species of xerothermic grasslands such as *Stipa joannis* or *Koeleria macrantha*, which occur in dry meadows (Figure 6). The more the taxon is to the left, the smaller the "light value" and the "higher the soil moisture value". In the left axis, there are such species as *Glyceria nemoralis* or *Calamagrostis canescens*, which are found in wet habitats. In relation to the vertical axis Y—PC2, the higher the taxon is, the smaller the "soil acidity value", which represents such species as *Festuca airoides* or *Calamagrostis villosa*, which are found in alpine tundras and forests, respectively, and the lower the taxon is, the higher the "soil acidity value", where we have species such as *Glyceria nemoralis* or *Calamagrostis pseudophragmites*. The most numerous group of taxa is in the middle and occurs simultaneously in different types of

habitats which have similar indicators. This analysis only shows the distribution of taxa in different habitats and indicates in which habitats the species can be found. However, it has not shown any correlation between the endangered species and their habitat requirements, as these species are found in different habitat types.

**Table 2.** Factor coordinates of the variables.

| Variables | Correlation | |
|---|---|---|
| | **PC1** | **PC2** |
| Light value (L) | 0.785565 | 0.274023 |
| Soil moisture value (F) | −0.77331 | −0.34218 |
| Soil acidity (pH) value (R) | 0.601228 | −0.79816 |

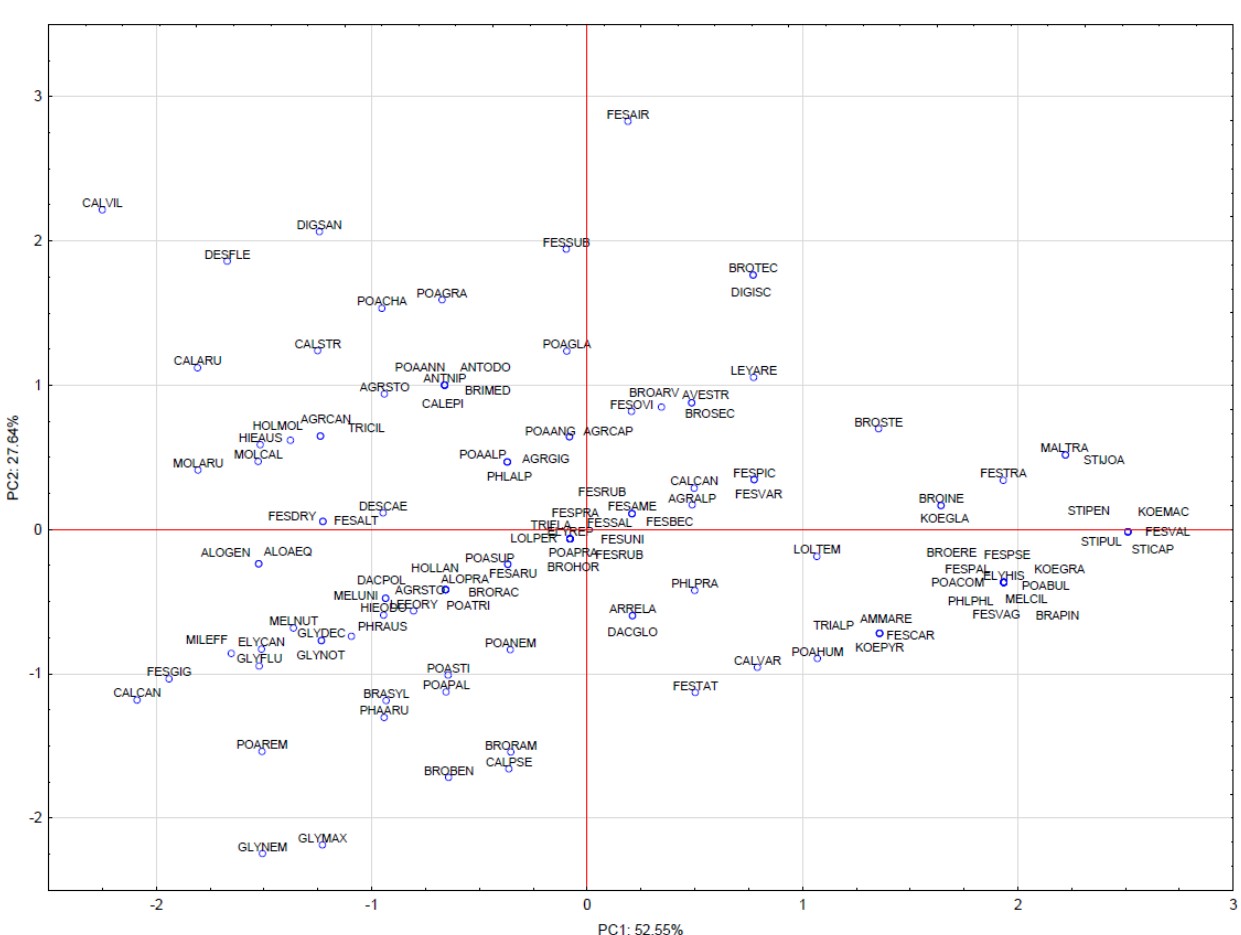

**Figure 6.** Principal components of analysis (PCA) of grasses according to the Ellenberg indicator values.

### 3.1. CWR in the Genetic Improvement of Other Species

The species of CWR from the *Poaceae* family are sources of genes unparalleled in modern varieties of crops (disease resistance, drought tolerance, etc.), which enable the extension of the genetic variability of, for example, wheat (*Triticum aestivum* L.). These species show homological similarity of chromosomes and have a high similarity of rDNA sequence to wheat. This enables the introgression of genes from grass species into the wheat genomes, where they can function permanently. For example, *Lolium perenne* was used to increase the winter hardiness of wheat [65].

For many years, breeders have been trying to create a perennial wheat, such as *Agropyron intermedium,* which was used as a donor for perennial growth [66,67]. However,

studies have so far provided evidence for substantial regrowth and perenniality for only two years.

The CWR species from the *Poaceae* family have a rather long history as a source of genes which can be transformed into a crop species such as wheat. The hybrids of wheat and *Agropyron* were described in 1933 [68] as to be of great scientific and practical interest. It has been postulated that through such hybrids, many new characteristics can be brought into utilization that presently do not exist in cereals. More recent research documented *Festuca arundinacea* as the main recipient in protoplast fusion, with common wheat as a donor [69]. The overall goal of such manipulations is to transfer a limited number of nuclear and plastid genes of wheat into this important pasture grass with the aim of widening its genetic base by introducing some important agronomic characters such as prolamins. In addition, the characteristics of the recipient, which has tetraploid and hexaploid status, may make the hybrids adaptable for a wide range of dosages of genetic materials.

Genetic plant resources, including landraces, have been collected and tested for adaptation and yield. The most suitable samples were multiplied, named, and often were improved by several cycles of mass selection of plants with greater pest resistance, better adaptation, etc. The next stage of improvement usually consists of developing synthetics produced by intermating several preferable clones, for yield, pest resistance, or drought tolerance. Many of the temperate grass cultivars have been improved and they are considered to be advanced generations of synthetics, involving very few clones. While strategies for grass improvement have focused primarily on intraspecific breeding in case of high agronomically desirable species *Lolium perenne* and *Lolium multiflorum,* a growing interest has emerged in interspecific hybrids as alternatives. × *Festulolium* refers to natural or synthetic intergeneric hybrids between obligate outbreeding species of the *Festuca* (fescue) and the *Lolium* (ryegrass) genera, which are considered frequently as ideal components of pasture or turf-grass systems. *Lolium* and *Festuca* species share complementary and desirable traits, and the prime aim in × *Festulolium* cultivar development has been to combine the agronomic performance of *Lolium* and the stress resistance of *Festuca* species. In the longer term, should our climates become consistently warmer and drier during the summer and/or liable to flooding due to extreme incidents of rainfall during autumn and winter, then their use may well increase [70].

It is visible that in the breeding of × *Festulolium* hybrids, significant progress in the global scale was achieved. Currently, 52 varieties of × *Festulolium* hybrids in the EU common catalogue are recorded, among which 5 are registered from Poland. From the point of view of the number of gained cultivars, the highest interest of breeding is focusing on *Festulolium braunii* hybrids obtained from *Lolium multiflorum* × *Festuca pratensis* crossing on the basis of amfiploidization and introgressive hybrids of *Festulolium pabulare* obtained from crossing between *Lolium multiflorum* (2x) × *Festuca arundinacea* var. *genuine* (6x). × *Festulolium* breeding has considerably stimulated research on genetics of the grasses and has contributed to the development of new technologies.

The practical implications of CWR species from the *Poaceae* family as elements in modern breeding programs has not yet been fully recognized or utilized. Reports of sterility are of much varied value. They are often a result of observation of single plants in conditions where a large abundance of gametes are needed for setting seeds. One such example is the work which has been conducted on the Polish hybrids of common oat and wild oat *Avena macrostachya*, where a grain that was able to sprout appeared in cloned and colchicined F1 generation in one of approximately 50–300 plants, depending on the hybrid genotype [71].

*3.2. The Potential of Crop Wild Relatives Direct Uses*

The CWR species from the *Poaceae* family have enormous direct uses in serving humans, animals, and the environment. Taking into the account the list of CWRs from the *Poaceae* family as presented in this paper, the following types of actual or potential species which can be considered in breeding were selected as follows:

### 3.2.1. Fodder and Edible Plants

Due to its specific phenotypic plasticity, different grass species can be used for diverse applications. The majority of them have great potential for animal feeding, whether bred by humans or in the wild. These species have major economic value, for example, *Lolium perenne, Dactylis glomerata, Festuca pratensis, F. arundinacea, F. rubra,* or *Bromus inermis.* It's probably that most of the CWR grass species could be more or less grazed by animals, and only few of them are unattractive or toxic. Health disorders were noted in horses, cows, sheep, or goats if they were fed with barley or oat contaminated with *Lolium temulentum* seeds [72]. In addition, when animals are fed with *Festuca arundinacea, F. pratensis,* or *Lolium perenne,* some health related problems may occur. The aforementioned species may live in symbiosis with endophytic fungus from the genus Neotyphodium, which is not visible over the whole stage of plant development. In certain conditions, endophytes may produce toxins harmful to animals such as ergovaline and lolitrem B [73]. The first clinical case of ergovaline intoxication in a herd of dairy cows in Poland was detected in 2018 [74]. Another type of application which was assigned was *Hierochloe odorata,* which is used as an aromatic additive in the Polish vodka "żubrówka". This grass species is distinguished by having a characteristic sweetish aroma with a hint of anise, which is due to it containing coumarin, which is especially strong after drying of the plant. It is probable that overexploitation of the natural habitats of the aforementioned species is the main reason why it is on Polska Czerwona Księga Roślin. Paprotniki i rośliny naczyniowe (=The Polish Red Data Book of Plants) [39].

### 3.2.2. Biomass Grasses

A number of perennial grasses are used increasingly in Europe and other parts of the world as renewable bioenergy sources, as they can grow with minimal maintenance on different kinds of soils and be harvested in large volumes of carbon-rich biomass [75]. If compared to energy produced from grain of corn or wheat, the utilization of grasses for the energy generation does not compete with the demand for consumable agricultural products [76]. Bioenergy crops are a typically densely populated, high yielding plant species. Over the last years, interest has focused on the cultivation of several groups of plants including perennial grasses; for example, C-4 grass species ad *Miscanthus* x *giganteus* and *Panicum virgatum* (not described thereafter) or native species from flora such as *Phalaris arundinacea* or *Festuca arundinacea.* However, currently there is no information concerning the wide application of the aforementioned native grass species for energy production in Poland.

Grass biomass, in addition to the energy generation, can be used to produce paper with satisfactory properties and employed in their use depending on the species which is used [77]. Pulps made from *Festuca arundinacea* and *Arrhenatherum elatius* biomass can be regarded as alternative sources of fibers that can be used to produce definite kinds of paper and cardboard products. Distinguished air permeability (>30 min) has been observed in the case of paper made from *Arrhenatherum elatius* biomass, as compared to paper from birch (19.4 s) or pine (12.6 s) [77].

In addition to being used for bioenergy purposes or paper pulp production, whole crops or residues of perennial grasses can be used as an addition in particleboard manufacturing. Our studies indicate that boards made with 30% additional grass biomass from *Festuca arundinacea* and *Lolium perenne* have acceptable mechanical properties and formaldehyde content less than the control particle board, i.e., a board made of 100% wood residues [78].

### 3.2.3. Amenity Grasses

'Amenity grasses' refers to their particular use, and is not related to food, forage, or bioenergy, but mostly to people's well-being, which according to the Cambridge Learners Dictionary, " . . . amenity means something that is intended or even necessary to make peoples life more pleasant or comfortable". Amenity grasses are quite unique among

cultivated crops due to so called phenotypic plasticity. The genotypes of lawn grass are capable of responding to fluctuating environments and stress by morphological and/or physiological changes. Even narrow leaf, dense, and close-cut genotypes may successfully survive due to their ability to produce seeds in noncompetitive conditions [79].

Species of the highest suitability for different lawn-related applications are *Agrostis capillaris*, *Lolium perenne*, *Festuca arudinacea*, and *Festuca rubra* ssp. *rubra.* The aforementioned species usually predominate and are suitable for many differing uses, i.e., golf greens and fairways, tennis and cricket fields, or polo and racing areas. In contrast, you can find species of a very limited usability such as *Poa compressa,* which is suitable only for the stabilization of areas of drought. It is most likely that the range of a species usability increases along with adaptation to a wide range of site conditions [79]. New and emerging species in lawn grass are *Deschampsia cespitosa* and *Koeleria macrantha*. Commercial varieties are currently accessible from Dutch seed companies, but our native genotypes have also proved their suitability for lawn conditions [80,81].

There are further uses for amenity grasses such as in ornamental grasses for landscape gardening (flower bed grasses) and floristry (species for dry flower arrangements such as *Briza media*, *Avena strigosa*, *Melica*, *Stipa*). Amenity grasses can be used as a specimen plant in perennial flower beds or are even better in large groupings and mass plantings (i.e., tall species; for example, *Deschampsia cespitiosa*, *Phalaris arundinacea*, and *Festuca gigantea*).

### 3.2.4. Ecology

The aforementioned category covers a range of species that grow in a wide range of soils, habitats, and microclimatic conditions. The first are a grass species sown in floral mixtures (grassy strips) for so called 'amenity landscaping' and 'habitat recreation'. For example: *Alopecurus pratensis*, *Anthoxantum odoratum*, *Briza media*, *Bromus erectus*, *Deschampsia flexuosa*, *Trisetum flavescens*, *Beckmannia eruciformis* [82].

Grass species from the CWR list are often used to remediate polluted soil. The species of confirmed suitability in the aforementioned application are: *Festuca arundinacea*, *F. pratensis*, *F. rubra*, *Dacylis glomerata*, *Bromus inermis*. It has been confirmed that *Festuca arundinacea* was able to absorb significantly higher amounts of $Cd^{2+}$ and $Zn^{2+}$ ions from polluted soil than other grass species, and it was not associated with yield decrease [83].

Some grass species are able to settle spontaneously on waste and remediated areas. Species such as *Dactylis glomerata*, *Calamagrostis epigejos*, *Agropyron repens*, *Poa pratensis*, and *Bromus mollis* were found to invade as a post-fire vegetation on remediated sulphur mine areas [84]. *Bromus tectorum* was found as a primary vegetation component on pure sand on the embankment of the tailings impoundment of "Żelazny Most" [85]. The aforementioned species outperformed the cultivated species (*Festuca arundinacea, F. rubra*) in terms of the ability to grow and develop on sand with high pH and high Ca contents [86]. Similar features determine the ability to stabilize extremely poor and arid habitats by *Ammophila arenaria*, *Calamagrostis epigejos*, and *Elymus arenarius* [87].

### 3.3. The Ex Situ Conservation in Gene Banks and Botanical Gardens

To develop new varieties, breeders need access to the genetic material of crop wild relatives, however, CWRs are not easily available, and therefore they need to access this material from the appropriate gene banks [88]. Safeguarding germplasm *ex situ* remains the most important approach to secure CWR biodiversity. Gene banks ought to provide a link between in situ (on farm) conservation and the breeders at various levels, because they are specialized in long-term preservation, distribution, and exchange of requested germplasm, evaluating the gathered accessions, keeping detailed databases on the individual accessions and, in some instances, conducting pre-breeding activities to facilitate the use of the germplasm [89]. The larger the *ex situ* collection in gene banks, the more material will be available to the breeder, but many CWR are missing from gene banks. In the central Polish gene bank in Radzików (NCPGR, National Centre for Plant Genetic Resources), all accessions are represented by approximately 30% of grasses germplasm [90,91], 10% of

which are CWR grasses. In the case of CWR grasses, 56% are stored in the NCPGR gene bank, and approximately 80% in botanical gardens (Appendix A). All taxa from this group of plants produce orthodox seeds and can be kept in long-term storage in gene banks, in which all collected and subsequently stored materials have their data time-stamped. Gene banks do not always have the valorization and evaluation of individual samples, which is an absolute requirement for plant breeders.

The potential availability of biological material has been checked using three international databases: GBIF—the Global Biodiversity Information Facility, Genesys—germplasm platform and EURISCO—The European Search Catalogue for Plant Genetic Resources. GBIF is an international network and data infrastructure that provides open access to data about all types of life on Earth (328,414,815 plant records). This knowledge derives from many sources, including everything from museum specimens collected in the 18th and 19th century to new DNA collections [92]. Genesys [93] possesses data on ex situ conserved germplasm accessions and provided data for 4,0997,112 accessions, 12% of which are classified as wild material [89]. EURISCO provides information about more than 2 million accessions of crop plants and their wild relatives, preserved *ex situ* by almost 400 institutes [94]. According to Ford-Lloyd et al. [19], the 1095 CWR species reported in EURISCO only represented 6% of the 17,495 CWR species found in Europe. This means that 94% of European CWR species are not conserved in *ex situ* collections [89]. In Poland, only around 23% of CWR taxa are stored in the Polish gene bank (NCPGR) [10]. However, in the case of CWR grasses, which constitute 10% of all CWR in Poland, 56% of them can be found in gene banks (Appendix A). The less common species are represented by a small number of accessions, which is a problem appointed by Engels and Thormann [89], among others.

The germplasm of 6 species of the genus *Festuca* was analyzed (Table 3). Common species, as well species of high economic importance and wide application, are represented in large numbers in gene bank collections (*F. rubra, F. arundincea*). It turned out that endemic species (*F. tatrae, F. macutrensis, F. carpatica*), due to the usually small number of sites, legal restrictions, etc., are poorly or not represented at all in genetic resource collections. It should be a task for both botanists and agricultural researches to fill the gaps. In addition, according to Engels and Thormann [89], "the application of new methods to assess the viability of seeds, not requiring seed germination test, could address current difficulties with viability tests and with small seed samples", which are a common problem with the conservation of CWR plants in gene banks. The continuation of collection missions carried out by specialists in order to collect the germplasm of rare end endemic species, which are represented by single accessions or not represented at all, is crucial to increase the CWR collection in gene banks.

**Table 3.** Occurrence of selected *Festuca* species in repositories *ex situ* on the Polish and international databases.

| Taxon Name | Occurrence (Number of Preserved Specimens) from GBIF [92] | Number of Accessions from EURISCO [93] | Number of Accessions from NCPGR [91] |
|---|---|---|---|
| *Festuca rubra* | 46,617 | 3079 | 471 |
| *F. arundinacea* | 7947 | 2872 | 586 |
| *F. gigantea* | 3640 | 236 | 6 |
| **_F. tatrae_** | 55 | 4 | 1 |
| **_F. macutrensis_** | 6 | 0 | 0 |
| **_F. carpatica_** | 92 | 1 | 0 |

Bold—endemic species in Poland.

Many botanical gardens hold living plant individuals in traditional display beds either indoors or outdoors, which serve as an important safety net for wild plant genetic resources [95,96]. In Poland, botanical gardens maintain an estimated 80% of Polish CWR grasses (Appendix A). However, it should be kept in mind that collections in botanical gardens are not permanent, and sometimes taxa are exchanged and replaced by others. Botanical gardens usually have few plants displayed, which are not a genetic representation of the populations in the wild. In contrast, they also have expertise in *ex situ* preservation, plant taxonomy, and horticulture, which are of paramount importance in breeding programs.

Considering both crop gene banks and botanical garden holdings result in a significantly higher number of conserved taxa, and together can maintain a wide range of *ex situ* collections useful for crop preservation, breeding, and research, while providing the necessary information and tools for plant breeders, the combined strengths and expertise within the crop gene bank and botanical garden communities make the *ex situ* preservation of all CWRs a goal that is within reach [95].

## 4. Conclusions

Summarizing, all the geographic and habitat elements of Polish flora are represented among CWR grass species. The *Poaceae* family could be considered an accurate model of the entire CWR flora in Poland and the adjacent countries. Amongst these grasses, Poland has representatives of many groups of cultivated and useful plants, which must be explored further. Despite a wide range of applications of CWR of grasses, it is still a very unexploited group of plants. However, it is only a matter of time before scientists and breeders will be able to tap into genetic information present in more than 140 species, in the hope of making other crop species more productive and resilient to extreme weather due to climate change. By studying the genetics of grass species closely related to major crops such as cereals, the researchers will be able to mine genes that encompass millions of years of evolutionary history. The gene bank should work closely with the botanical gardens to ensure full *ex situ* preservation of all CWR taxa in the country.

**Author Contributions:** Conceptualization, D.F.D.; methodology, D.F.D., A.K.; validation, D.F.D., A.K., G.Ż. and W.P.; formal analysis, D.F.D.; investigation, D.F.D., A.K..; data curation, D.F.D.; writing—original draft preparation, D.F.D., A.K., G.Ż. and W.P.; writing—review and editing, D.F.D.; visualization, D.F.D.; supervision, D.F.D.; funding acquisition, D.F.D. and W.P. All authors have read and agreed to the published version of the manuscript.

**Funding:** This work was supported by the Multi-annual program: 2015–2020 "Establishment of a scientific basis for biological progress and preservation of plant genetic resources as a source of innovation in order to support sustainable agriculture and food security of the country" coordinated by Plant Breeding and Acclimatization Institute (IHAR)-National Research Institute and financed by the Ministry of Agriculture and Rural Development of Poland.

**Institutional Review Board Statement:** Not applicable.

**Informed Consent Statement:** Not applicable.

**Data Availability Statement:** The authors obtained permission to use the ATPOL database.

**Acknowledgments:** For Adam Zając, for sharing the ATPOL database in order to prepare map compilations.

**Conflicts of Interest:** The authors declare no conflict of interest.

## Appendix A

**Table A1.** Checklist of crop wild relatives among grasses (Poaceae) occurring in Poland.

| Taxon Name | Taxa Acronym | Floristic Status in Polish Flora | Potential Direct Use | | | | | Preferred Habitats | European Red List (Vascular Plants/Medicinal Plants) | Polish Red List/Polish Red Data Book | Ellenberg's Indicator Values | | | Presence in Collections | | | |
|---|---|---|---|---|---|---|---|---|---|---|---|---|---|---|---|---|---|
| | | | Fodder | Edible | Amenity Grasses | Biomass, Fuels and Raw Material | Ecology | | | | Light Value (L) | Soil Moisture Value (M) | Soil Acidity (pH) Value (R) | Botanical Gardens and Arboretum | NCPGR | KFGB | Seed Banks |
| *Agrostis alpina* Scop. | AGRALP | N | • | | | | | 10, 11 | | VU/VU | 5 | 3–4 | 3–5 | | | | |
| *Agrostis canina* L. | AGRCAN | N | • | | • | | | 4 | LC/- | | 4 | 4 | 3 | • | • | | |
| *Agrostis capillaris* L. | AGRCAP | N | • | | | | | 10 | | | 4 | 2–3 | 3–4 | • | • | • | |
| *Agrostis gigantea* Roth | AGRGIG | N | • | | • | | | 2 | | | 4 | 3 | 3–4 | • | • | | |
| *Agrostis rupestris* All. | AGRRUP | N | • | | | | | 10 | | | | | | • | • | | |
| *Agrostis stolonifera* L. | AGRSTO | N | • | | • | | | 2 | LC/- | | 4 | 4 | 3–5 | • | • | | |
| *Agrostis vinealis* Schreb. | AGRSTO | N | • | | | | | 3 | | DD/- | 3 | 2 | 2–4 | • | | | |
| *Alopecurus aequalis* Sobol. | ALOAEQ | N | • | | | | | 2 | LC/- | | 4 | 5 | 3–4 | • | | | |
| *Alopecurus arundinaceus* Poir. In Lam. | ALOARU | N | • | | | | • | 2, 10, 11, 12 | | | | | | • | | | |
| *Alopecurus geniculatus* L. | ALOGEN | N | • | | | | | 2 | LC/- | | 4 | 5 | 3–4 | • | | | |
| *Alopecurus pratensis* L. | ALOPRA | N | • | | • | | • | 2 | LC/- | | 4 | 4 | 4 | • | • | | |
| *Ammophila arenaria* (L.) Link | AMMARE | N | | • | • | | • | 13 | | | 5 | 3 | 5 | • | • | | |
| *Anthoxanthum odoratum* L. | ANTODO | N | • | • | | | | 3, 4, 5, 6 | | | 4 | 3 | 3 | • | • | | |

Table A1. *Cont.*

| Taxon Name | Taxa Acronym | Floristic Status in Polish Flora | Potential Direct Use | | | | | Preferred Habitats | European Red List (Vascular Plants/Medicinal Plants) | Polish Red List/Polish Red Data Book | Ellenberg's Indicator Values | | | Presence in Collections | Presence in Gene Banks | | |
| | | | Fodder | Edible | Amenity Grasses | Biomass, Fuels and Raw Material | Ecology | | | | Light Value (L) | Soil Moisture Value (M) | Soil Acidity (pH) Value (R) | Botanical Gardens and Arboretum | NCPGR | KFGB | Seed Banks |
|---|---|---|---|---|---|---|---|---|---|---|---|---|---|---|---|---|---|
| *Anthoxanthum odoratum* subsp. *nipponicum* (Honda) Tzvelev | ANTNIP | N | • | • | | | • | 10, 11 | | | 4 | 3 | 3 | | | | |
| *Arrhenatherum elatius* (L.) P. Beauv. ex J. Presl & C. Presl | ARRELA | N | • | • | • | • | • | 2, 3 | LC/- | | 4 | 3 | 4–5 | | • | • | |
| *Avena strigosa* Schreb. | AVESTR | A | • | • | • | | | 4 | | DD/- | 5 | 3 | 3–4 | | • | • | |
| *Beckmannia eruciformis* (L.) Host | BECERU | A | • | | • | | • | 2 | | | | | | | | | |
| *Brachypodium pinnatum* (L.) P. Beauv. | BRAPIN | N | | • | | | • | 1, 6 | | | 5 | 2 | 5 | | • | • | |
| *Brachypodium sylvaticum* (Huds.) P. Beauv. | BRASYL | N | | • | | | | 5, 6 | | | 3 | 3–4 | 4–5 | | • | • | |
| *Briza media* L. | BRIMED | N | | • | • | | | 2 | | | 4 | 3 | 2–4 | | • | • | |
| *Bromus arvensis* L. | BROARV | A | • | • | | | | 4 | | VU/- | 4–5 | 2–3 | 3–4 | | • | | |
| *Bromus benekenii* (Lange) Trimen | BROBEN | N | • | • | | | | 5, 6 | | | 3 | 3–4 | 5 | | • | • | |
| *Bromus commutatus* Schrad. | BROCUM | N | • | • | | | | | | | | | | | | | |
| *Bromus erectus* Huds. | BROERE | N | • | • | | | • | 1 | | | 5 | 2 | 5 | | • | • | |
| *Bromus hordeaceus* L. | BROHOR | N | | • | | | | 2 | | | 4 | 3 | 4 | | • | • | |
| *Bromus inermis* Leyss. | BROINE | N | • | • | | • | • | 1 | | | 5 | 2 | 4–5 | | • | • | |

Table A1. *Cont.*

| Taxon Name | Taxa Acronym | Floristic Status in Polish Flora | Potential Direct Use | | | | | Preferred Habitats | European Red List (Vascular Plants/Medicinal Plants) | Polish Red List/Polish Red Data Book | Ellenberg's Indicator Values | | | Presence in Collections | | | |
| | | | Fodder | Edible | Amenity Grasses | Biomass, Fuels and Raw Material | Ecology | | | | Light Value (L) | Soil Moisture Value (M) | Soil Acidity (pH) Value (R) | Botanical Gardens and Arboretum | Presence in Gene Banks | | |
| | | | | | | | | | | | | | | | NCPGR | KFGB | Seed Banks |
|---|---|---|---|---|---|---|---|---|---|---|---|---|---|---|---|---|---|
| *Bromus racemosus* L. | BRORAC | N | • | • | | | | 1 | | NT/- | 4 | 4 | 4 | | | | |
| *Bromus ramosus* Huds. | BRORAM | N | | • | | | | 5, 6 | | VU/- | 3 | 3 | 5 | | • | • | |
| *Bromus secalinus* L. | BROSEC | A | | • | • | | | 4 | | | 5 | 3 | 3–4 | | • | • | • |
| *Bromus sterilis* L. | BROSTE | A | | | | | • | 4 | | | 5 | 2 | 4 | | • | • | |
| *Bromus tectorum* L. | BROTEC | A | | • | | | • | 4 | | | 5 | 2 | 3 | | • | • | |
| *Calamagrostis arundinacea* (L.) ROTH | CALARU | N | | | | • | • | 5, 6, 7, 8 | | | 3 | 3 | 2–3 | | • | • | |
| *Calamagrostis canescens* (Weber) Roth | CALCAN | N | • | | | | • | 2, 5 | | | 3 | 5 | 4 | | | • | |
| *Calamagrostis epigejos* (L.) Roth | CALEPI | N | | | | | • | 4 | | | 4 | 3 | 3 | | • | • | |
| *Calamagrostis pseudophragmites* (Haller F.) Koeler | CALPSE | N | • | | | | • | 2 | | | 4 | 4–5 | 5 | | • | | |
| *Calamagrostis stricta* (Timm) Koeler | CALSTR | N | • | | | • | | 2 | | NT/- | 5 | 5 | 2–3 | | • | | |
| *Calamagrostis varia* (Schrad.) Host | CALVAR | N | • | | | • | | 8, 11 | | | 4 | 2–3 | 5 | | • | | |
| *Calamagrostis villosa* (Chaix) J. F. Gmel. | CALVIL | N | • | | | • | | 5 | | | 4–3 | 3–4 | 1–2 | | • | | |

Table A1. *Cont.*

| Taxon Name | Taxa Acronym | Floristic Status in Polish Flora | Fodder | Edible | Amenity Grasses | Biomass, Fuels and Raw Material | Ecology | Preferred Habitats | European Red List (Vascular Plants/Medicinal Plants) | Polish Red List/Polish Red Data Book | Light Value (L) | Soil Moisture Value (M) | Soil Acidity (pH) Value (R) | Botanical Gardens and Arboretum | NCPGR | KFGB | Seed Banks |
|---|---|---|---|---|---|---|---|---|---|---|---|---|---|---|---|---|---|
| *Corynephorus canescens* (L.) P. Beauv. | CALCAN | N | | | | | • | 9, 13 | | | 4 | 2 | 3–5 | • | • | | |
| *Dactylis glomerata* L. * | DACGLO | N | • | • | | • | | 2,3 | | | 4 | 3 | 4–5 | • | • | | • |
| *Dactylis polygama* Horv. * | DACPOL | N | • | • | | | | 5, 6 | | | 3 | 3 | 4 | • | | | • |
| *Deschampsia caespitosa* (L.) P. Beauv. | DESCAE | N | | • | • | • | • | 2 | | | 3–5 | 4 | 3–4 | • | • | | |
| *Deschampsia flexuosa* (L.) Trin. | DESFLE | N | | • | | | | 5, 6, 7, 8 | | | 3–4 | 3 | 1–3 | • | • | | |
| *Deschampsia setacea* (Huds.) Hack. | DESSET | E | | • | | | | 2 | | RE/EX | | | | | | | |
| *Digitaria ischaemum* (Schreb.) H. L. Mühl. | DIGISC | A | | • | | | | 4 | | | 5 | 2 | 3 | • | • | | |
| *Digitaria sanguinalis* (L.) Scop. | DIGSAN | A | | • | | | | 4 | | | 4 | 3 | 2 | • | • | | |
| *Elymus caninus* (L.) L. | ELYCAN | N | | • | | | | 4 | LC/- | | 3 | 4 | 4 | • | | | |
| *Elymus farctus* (Viv.) Runemark ex Melderis * | ELYFAR | N | | • | | | | 13 | | -/CR | | | | | • | • | • |
| *Elymus hispidus* (Opiz) Melderis | ELYHIS | N | | • | | | | 4 | | | 5 | 2 | 5 | • | • | | |
| *Elymus repens* (L.) Gould | ELYREP | N | • | • | | | | 4 | | | 4 | 3 | 3–5 | • | • | | |
| *Festuca airoides* Lam. | FESAIR | N | • | | | | | 10 | | | 5 | 2 | 2 | • | | | |
| *Festuca altissima* All. | FESALT | N | | • | • | | • | 5, 6 | | | 3 | 3 | 3–4 | • | • | | |

**Table A1.** *Cont.*

| Taxon Name | Taxa Acronym | Floristic Status in Polish Flora | Potential Direct Use | | | | | Preferred Habitats | European Red List (Vascular Plants/Medicinal Plants) | Polish Red List/Polish Red Data Book | Ellenberg's Indicator Values | | | Presence in Collections | Presence in Gene Banks | | |
|---|---|---|---|---|---|---|---|---|---|---|---|---|---|---|---|---|---|
| | | | Fodder | Edible | Amenity Grasses | Biomass, Fuels and Raw Material | Ecology | | | | Light Value (L) | Soil Moisture Value (M) | Soil Acidity (pH) Value (R) | Botanical Gardens and Arboretum | NCPGR | KFGB | Seed Banks |
| *Festuca amethystina* L. | FESAME | N | | | ● | | ● | 10, 11 | | EN/EN | 4 | 2–3 | 4 | ● | | | |
| *Festuca amethystina* ssp. *ritschlii* * | FESRIT | N | | | ● | | ● | 5, 6 | | | | | | | | | |
| *Festuca arundinacea* Schreb. | FESARU | N | ● | | ● | ● | ● | 2, 12 | | | 4 | 3–4 | 4 | ● | ● | | |
| **Festuca beckeri (Hack.) Trautv.** | FESBEC | N | ● | | | | | 3 | | NT/- | 4 | 2–3 | 5 | ● | ● | | |
| **Festuca carpathica F.Dietr.** | FESCAR | N | ● | | | | | 11 | | | 5 | 3 | 5 | ● | | | |
| *Festuca diffusa* Dumort. | FESDIF | N | ● | | | | | 2 | | | | | | | | | |
| *Festuca drymeia* Mert. & W. D. J. Koch | FESDRY | N | | ● | ● | | | 2 | | | 3 | 3 | 3–4 | | | | |
| *Festuca duriuscula* L. *Festuca rubra* L. | FESRUB | N | ● | | ● | | ● | 2,3 | | | 4 | 2–4 | 5–6 | | | | |
| *Festuca duvalii* (St. Yves) Stohr | FESDUV | N | ● | | | | | 3 | | DD/- | | | | | | | |
| *Festuca filiformis* Pourr. | FESFIL | N | ● | | ● | | | 2 | | DD/- | 4 | 3–4 | | | ● | ● | |
| *Festuca gigantea* (L.) Vill. | FESGIG | N | | ● | | | | 5 | | | 2–3 | 4 | 4 | ● | ● | | |
| *Festuca heterophylla* Lam. | FESHET | N | ● | | ● | | | 6 | LC/- | NT/- | | | | | ● | ● | |
| **Festuca macutrensis Zapał.** | FESMAC | N | ● | | | | | 1 | | EN/EN | 2–3 | 4 | 5–6 | | | | |
| *Festuca nigrescens* Lam. | FESNIG | N | ● | | ● | | ● | 2 | | DD/ | | | | | ● | ● | |

Table A1. *Cont.*

| Taxon Name | Taxa Acronym | Floristic Status in Polish Flora | Potential Direct Use | | | | | Preferred Habitats | European Red List (Vascular Plants/Medicinal Plants) | Polish Red List/Polish Red Data Book | Ellenberg's Indicator Values | | | Presence in Collections | | | |
| | | | Fodder | Edible | Amenity Grasses | Biomass, Fuels and Raw Material | Ecology | | | | Light Value (L) | Soil Moisture Value (M) | Soil Acidity (pH) Value (R) | Botanical Gardens and Arboretum | Presence in Gene Banks | | |
| | | | | | | | | | | | | | | | NCPGR | KFGB | Seed Banks |
| *Festuca ovina* L. * | FESOVI | N | ● | | ● | | ● | 3 | LC/- | | 4 | 2 | 3–4 | ● | ● | | |
| *Festuca ovina* L. var. *vulgaris* W.D.J.Koch subvar. *guestphalica* Hack. * | FESGUE | N | ● | | ● | | | 3 | | DD/- | 4 | 2 | 5–6 | | | | |
| *Festuca pallens* Host | FESPAL | N | | | ● | | | 11 | | | 5 | 2 | 5 | ● | ● | | |
| *Festuca picta* Kit. * | FESPIC | N | | | | | ● | 11 | | | 5 | 3 | 4 | ● | | | |
| *Festuca pratensis* Huds. | FESPRA | N | ● | | | ● | ● | 2, 3 | | | 4 | 3 | 4 | ● | ● | | |
| *Festuca psammophila* (Hack. ex Celak.) Fritsch | FESPSA | N | ● | | ● | | | 3 | NT/- | | 5 | 2 | | ● | | | |
| *Festuca pseudodalmatica* Krajina ex Domin | FESDAL | N | ● | | ● | | | 1 | | CR/CR | | | | ● | | | |
| *Festuca pseudovina* Hack. ex Wiesb. | FESPSE | N | ● | | ● | | | 3 | | CR/CR | 5 | 2 | 5 | ● | | | |
| *Festuca rubra* L. s. str. * | FESRUB | N | ● | | ● | | ● | 2 | LC/- | | 4 | 2–4 | 4–6 | ● | ● | | |
| *Festuca rupicola* Heuff. | FESRUP | N | ● | | ● | | | 1 | | | 5 | | 5 | ● | | | |
| *Festuca salina* Natho & Stohr * | FESSAL | N | ● | | | | | 12 | | | 4 | 2–4 | 4 | | | | |
| ***Festuca tatrae* (Czakó) Degen** | FESTAT | N | ● | | ● | | | 10, 11 | | | 4 | 3 | 5 | ● | ● | | |
| *Festuca trachyphylla* (Hack.) Krajina | FESTRA | N | ● | | ● | | | 1 | | | 5 | 1–2 | 5–6 | ● | | | |

Table A1. *Cont.*

| Taxon Name | Taxa Acronym | Floristic Status in Polish Flora | Potential Direct Use | | | | | Preferred Habitats | European Red List (Vascular Plants/Medicinal Plants) | Polish Red List/Polish Red Data Book | Ellenberg's Indicator Values | | | Presence in Collections | Presence in Gene Banks | | |
| | | | Fodder | Edible | Amenity Grasses | Biomass, Fuels and Raw Material | Ecology | | | | Light Value (L) | Soil Moisture Value (M) | Soil Acidity (pH) Value (R) | Botanical Gardens and Arboretum | NCPGR | KFGB | Seed Banks |
|---|---|---|---|---|---|---|---|---|---|---|---|---|---|---|---|---|---|
| *Festuca unifaria* Dumort. * | FESUNI | N | ● | | ● | | ● | 2 | | | 4 | 2–4 | 4 | | | | |
| *Festuca vaginata* Waldst. & Kit. ex Willd. | FESVAG | N | ● | | ● | | | 2 | | DD/- | 5 | 2 | 5 | | ● | ● | |
| *Festuca valesiaca* Schleich. ex Gaudin | FESVAL | N | | | ● | | | 1 | | VU/- | 5 | 1 | 5 | | ● | ● | |
| *Festuca varia* Haenke | FESVAR | N | ● | | ● | | | 11 | | | 5 | 3 | 4 | ● | | | |
| *Festuca villosa* Schweigg. | FESVIL | N | ● | | ● | | | | | | | | | | | | |
| *Glyceria declinata* Bréb. | GLYDEC | N | ● | | | | | 2 | LC/- | | 4 | 5 | 4 | | | | |
| *Glyceria fluitans* (L.) R. Br. | GLYFLU | N | | ● | | | | 2 | LC/- | | 4 | 6–5 | 4 | | ● | ● | |
| *Glyceria lithuanica* (Gorski) Gorski | GLYLIT | N | ● | | | | | 7 | | CR/CR | | | | | ● | | |
| *Glyceria maxima* (Hartm.) Holmb. | GLYMAX | N | ● | | ● | | ● | 2 | LC/- | | 4 | 6 | 5 | | ● | ● | |
| *Glyceria nemoralis* (R. Uechtr.) R. Uechtr. & Körn. | GLYNEM | N | | ● | | | | 6 | LC/- | | 3 | 5 | 5 | | | | |
| *Glyceria notata* Chevall. | GLYNOT | N | | ● | | | | 2 | | | 4 | 5 | 4 | ● | | | |
| *Glyceria striata* (Lam.) Hitchc. | GLYSTR | A | | | ● | | | 2 | | | | | | | | | |
| *Hierochloë australis* (Schrad.) Roem. & Schult. | HIEAUS | N | | ● | ● | | ● | 5, 6, 7, 8 | | VU/- | 3 | 3 | 3 | ● | | | ● |

Table A1. *Cont.*

| Taxon Name | Taxa Acronym | Floristic Status in Polish Flora | Potential Direct Use | | | | | Preferred Habitats | European Red List (Vascular Plants/Medicinal Plants) | Polish Red List/Polish Red Data Book | Ellenberg's Indicator Values | | | Presence in Collections | Presence in Gene Banks | | |
| --- | --- | --- | --- | --- | --- | --- | --- | --- | --- | --- | --- | --- | --- | --- | --- | --- | --- |
| | | | Fodder | Edible | Amenity Grasses | Biomass, Fuels and Raw Material | Ecology | | | | Light Value (L) | Soil Moisture Value (M) | Soil Acidity (pH) Value (R) | Botanical Gardens and Arboretum | NCPGR | KFGB | Seed Banks |
| *Hierochloë odorata* (L.)P. Beauv. | HIEODO | N | | ● | | | | 1,3, 5, 6 | | VU/- | 4 | 4–5 | 4 | ● | | ● | |
| *Holcus lanatus* L. | HOLLAN | N | ● | | | | | 2 | | | 4 | 4 | 4 | ● | ● | | |
| *Holcus mollis* L. | HOLMOL | N | | | | | ● | 9 | | | 3–4 | 3–4 | 3 | ● | | | |
| *Koeleria glauca*(Spreng.) Dc. | KOEGLA | N | | | ● | | | 3 | | | 5 | 2 | 4–5 | ● | ● | | |
| *Koeleria grandis* Besser ex Gorski * | KOEGRA | N | | | ● | | | 1 | | DD/- | 5 | 2 | 5 | ● | ● | | |
| *Koeleria macrantha* (Ledeb.) Schult. | KOEMAC | N | | | ● | | ● | 3 | | | 5 | 1 | 5 | ● | ● | | ● |
| *Koeleria pyramidata* (Lam.) P. Beauv. * | KOEPYR | N | | | ● | | ● | 1 | | VU/- | 5 | 3 | 5 | ● | ● | | |
| *Leersia oryzoides* (L.) Sw. | LEEORY | N | | ● | | | | 2 | LC/- | NT/- | 5–4 | 5 | 4 | | ● | | |
| *Leymus arenarius (L.)* Hochst. * | LEYARE | N | | ● | ● | ● | ● | 13 | | | 5 | 2–3 | 3–4 | ● | ● | | |
| *Lolium perenne* L. | LOLPER | N | ● | | ● | ● | | 4 | LC/- | | 4 | 3 | 4 | ● | ● | | |
| *Lolium remotum* Schrank | LOLREM | A | ● | | | | | 4 | | CR/- | | | | | ● | ● | |
| *Lolium temulentum* L. | LOLTEM | A | ● | | | | | 4 | LC/- | VU/- | 5 | 3 | 4–5 | ● | ● | | |
| *Melica ciliata* L. | MELCIL | N | | | ● | | | 1 | | | 5 | 2 | 5 | ● | ● | | ● |
| *Melica nutans* L. | MELNUT | N | ● | ● | ● | | | 5, 6 | | | 2–3 | 3 | 4 | ● | ● | | |
| *Melica picta* K. Koch | MELPIC | N | | ● | ● | | | 5, 6 | | CR/CR | | | | | ● | | |

**Table A1.** *Cont.*

| Taxon Name | Taxa Acronym | Floristic Status in Polish Flora | Potential Direct Use | | | | | Preferred Habitats | European Red List (Vascular Plants/Medicinal Plants) | Polish Red List/Polish Red Data Book | Ellenberg's Indicator Values | | | Presence in Collections | | | |
|---|---|---|---|---|---|---|---|---|---|---|---|---|---|---|---|---|---|
| | | | Fodder | Edible | Amenity Grasses | Biomass, Fuels and Raw Material | Ecology | | | | Light Value (L) | Soil Moisture Value (M) | Soil Acidity (pH) Value (R) | Botanical Gardens and Arboretum | Presence in Gene Banks | | |
| | | | | | | | | | | | | | | | NCPGR | KFGB | Seed Banks |
| *Melica transsilvanica* **Schur** | MALTRA | N | | | ● | | | 1 | | NT/- | 5 | 1 | 4–5 | ● | ● | ● | |
| *Melica uniflora* Retz. | MELUNI | N | | ● | ● | | | 5, 6 | | | 3 | 3 | 4 | ● | | | |
| *Milium effusum* L. | MILEFF | N | ● | ● | | | | 5, 6 | | | 2–3 | 3–4 | 4 | ● | ● | | |
| *Molinia arundinacea* Schrank * | MOLARU | N | | ● | ● | | | 2 | | | 3 | 3–4 | 3 | ● | ● | | |
| *Molinia caerulea* (L.) Moench * | MOLCAL | N | | ● | ● | | | 2 | | | 4 | 4–5 | 1–5 | ● | ● | | |
| *Phalaris arundinacea* L. | PHAARU | N | ● | ● | ● | ● | ● | 2 | LC/- | | 4 | 5 | 4–5 | ● | ● | | |
| *Phleum alpinum* L. | PHLALP | N | ● | | | | | 10, 11 | | | 4 | 3 | 3–4 | ● | ● | | |
| *Phleum bertolonii* Dc. * (*P. hubbardii* D. Kovats) | PHLBER | N | ● | ● | ● | | | 3 | | | 4 | 2 | 5–6 | | | | |
| *Phleum hirsutum* Honck. | PHLHIR | N | ● | ● | | | | 10, 11 | | | | | | | | | |
| *Phleum phleoides* (L.) H. Karst. | PHLPHL | N | | ● | | | | 2 | | | 5 | 2 | 5 | ● | ● | | ● |
| *Phleum pratense* L. | PHLPRA | N | ● | ● | | ● | | 2, 3 | LC/- | | 4 | 2–3 | 4–5 | ● | ● | | |
| *Phragmites australis* (Cav.) Trin. ex Steud. | PHRAUS | N | | ● | ● | ● | ● | 12 | LC/- | | 4–5 | 5–6 | 4 | ● | | | |

**Table A1.** *Cont.*

| Taxon Name | Taxa Acronym | Floristic Status in Polish Flora | Potential Direct Use | | | | | Preferred Habitats | European Red List (Vascular Plants/Medicinal Plants) | Polish Red List/Polish Red Data Book | Ellenberg's Indicator Values | | | Presence in Collections | | | |
|---|---|---|---|---|---|---|---|---|---|---|---|---|---|---|---|---|---|
| | | | Fodder | Edible | Amenity Grasses | Biomass, Fuels and Raw Material | Ecology | | | | Light Value (L) | Soil Moisture Value (M) | Soil Acidity (pH) Value (R) | Botanical Gardens and Arboretum | Presence in Gene Banks | | |
| | | | | | | | | | | | | | | | NCPGR | KFGB | Seed Banks |
| *Poa alpina* L. | POAALP | N | ● | | | | | 10, 11 | LC/- | | 4 | 3 | 3–4 | ● | ● | | |
| *Poa angustifolia* L. | POAANG | N | ● | | | | | 10, 11 | | | 4 | 2–3 | 4–5 | | ● | | |
| *Poa annua* L. | POAANN | N | | | ● | | | 4 | | | 5–3 | 3 | 4 | ● | | | |
| *Poa bulbosa* L. | POABUL | N | ● | | | | | 1 | NT/- | | 5 | 2 | 5 | ● | ● | | |
| *Poa chaixii* Vill. | POACHA | N | ● | | | | | 10, 11 | | | 4 | 3 | 2–3 | ● | ● | | |
| *Poa compressa* L. | POACOM | N | ● | | ● | | ● | 1 | | | 5 | 2 | 5 | ● | ● | | |
| *Poa glauca* Vahl | POAGLA | N | ● | | ● | | | 11 | | | 5 | 3–4 | 3 | ● | ● | | |
| ***Poa granitica* Braun–Blanq.** | POAGRA | N | ● | | | | | 10 | DD/- | NT/NT | 5 | 4 | 2–3 | | | | |
| *Poa humilis* Ehrh. ex Hoffm. | POAHUM | N | ● | | | | | 1, 2, 13 | | | 5 | 4–3 | 5 | | | | |
| *Poa laxa* Haenke | FESSUB | N | ● | | ● | | | 10 | | | 5 | 3 | 2–3 | ● | | | |
| *Poa molinerii* Balb. | POAMOL | N | ● | | | | | 10, 11 | | | 5 | 3 | 2–3 | | | | |
| *Poa nemoralis* L. | POANEM | N | ● | | ● | | | 5, 6 | | | 3 | 2–3 | 4–5 | ● | ● | | |
| ***Poa nobilis* Skalińska** | PONNOB | N | ● | | | | | 10 | | DD/DD | | | | | | | |
| *Poa palustris* L. | POAPAL | N | ● | | ● | | | 2 | | | 4 | 4–5 | 4–5 | ● | ● | ● | |

**Table A1.** *Cont.*

| Taxon Name | Taxa Acronym | Floristic Status in Polish Flora | Potential Direct Use | | | | | Preferred Habitats | European Red List (Vascular Plants/Medicinal Plants) | Polish Red List/Polish Red Data Book | Ellenberg's Indicator Values | | | Presence in Collections | | | |
|---|---|---|---|---|---|---|---|---|---|---|---|---|---|---|---|---|---|
| | | | Fodder | Edible | Amenity Grasses | Biomass, Fuels and Raw Material | Ecology | | | | Light Value (L) | Soil Moisture Value (M) | Soil Acidity (pH) Value (R) | Botanical Gardens and Arboretum | Presence in Gene Banks | | |
| | | | | | | | | | | | | | | | NCPGR | KFGB | Seed Banks |
| *Poa pratensis* L. | POAPRA | N | ● | | ● | | | 2 | LC/- | | 4 | 3 | 4 | ● | ● | | |
| *Poa remota* Forselles | POAREM | N | ● | | | | | 10, 11 | | | 3 | 4–5 | 4–5 | | | | |
| *Poa stiriaca* Fritsch & Hayek | POASTI | N | ● | | | | | 6 | | VU/VU | 3 | 3 | 4–5 | ● | ● | | |
| *Poa supina* Schrad. | POASUP | N | ● | | ● | | | 10 | | | 4 | 3–4 | 4 | ● | | | |
| *Poa trivialis* L. | POATRI | N | ● | | | | | 2 | | | 4 | 4 | 4 | ● | ● | | |
| *Stipa capillata* L. | STICAP | N | | ● | ● | | | 1 | | VU/- | 5 | 1 | 5 | ● | ● | | ● |
| *Stipa eriocaulis* Borbás * | STIEUR | N | | | ● | | | 1 | | RE/- | | | | | ● | | |
| ***Stipa joannis* Čelak. s. s.** | STIJOA | N | | ● | ● | | | 1 | | VU/VU | 5 | 1 | 5–4 | ● | | ● | ● |
| *Stipa pennata* L. (Stipa borysthenica Klokov) * | STIPEN | N | | ● | ● | | | 1 | | CR/CR | 5 | 1 | 5 | ● | | | ● |
| *Stipa pulcherrima* K. Koch | STIPUL | N | | ● | ● | | | 1 | | VU/VU | 5 | 1 | 5 | ● | | ● | ● |
| *Trisetum alpestre* (Host) P. Beauv. | TRIALP | N | ● | | | | | 10, 11 | | | 5 | 3 | 5 | ● | | | |
| *Trisetum ciliare* (Kit.) Domin | TRICIL | N | ● | | | | | 10, 11 | | | 4 | 4 | 3 | | | | |

Table A1. *Cont.*

| Taxon Name | Taxa Acronym | Floristic Status in Polish Flora | Potential Direct Use | | | | | Preferred Habitats | European Red List (Vascular Plants/Medicinal Plants) | Polish Red List/Polish Red Data Book | Ellenberg's Indicator Values | | | Presence in Collections | | | |
|---|---|---|---|---|---|---|---|---|---|---|---|---|---|---|---|---|---|
| | | | Fodder | Edible | Amenity Grasses | Biomass, Fuels and Raw Material | Ecology | | | | Light Value (L) | Soil Moisture Value (M) | Soil Acidity (pH) Value (R) | Botanical Gardens and Arboretum | Presence in Gene Banks | | |
| | | | | | | | | | | | | | | | NCPGR | KFGB | Seed Banks |
| *Trisetum flavescens* (L.) P. Beauv. | TRIFLA | N | ● | | | | ● | 11 | | | 4 | 3 | 4 | ● | ● | | |
| *Trisetum sibiricum* Rupr. | TRISIB | N | ● | | | | | 2 | NT/NT | | 4 | 4 | | | | | |
| Total | | | | | | | | | | | | | | 116 | 82 | 10 | 8 |

Explanations: This appendix was prepared based on work presented in the "Dzikie gatunki pokrewne roślinom uprawnym występujące w Polsce. Lista, zasoby i zagrożenia" (*Crop wild relatives occurring in Poland. Checklist, resources and threats*) [10]. Latin name—Flowering plants and pteridophytes of Poland—A checklist [29]/The plant List [30]. Species name in bold—endemic species in Poland. * The taxonomy name has been accepted just according to the Flowering plants and pteridophytes of Poland—A checklist [29], because it includes several endemic species that are widely accepted by botanists and geographers from all over Central Europe (Germany, Czech Republic, Slovakia, Hungary, Belarus, Lituania), not only Poland, as stenochoric, but distinct species. Most of them are Tertiary or Pleistocene Carpathian endemics. Taxa acronym: 3 letters of genera and 3 letters of the epithet. Status in Polish Flora—N: Native; A: Archeophyte, E: Extinct, according to: Flowering plants and pteridophytes of Poland—A checklist [29]/Rośliny obcego pochodzenia w Polsce [33]. Preferred habitat: 1—steppes and xerothermous grasslands, 2—wet meadows and pastures, 3—dry meadows and pastures, 4—arable fields and balks, 5—wet deciduous forests, 6—dry deciduous forests, 7—wet coniferous forests, 8—dry coniferous forests, 9—heath, 10—alpine tundra on granite and similar acidic rock substrate, 11—alpine tundra on limestone and similar alkaline rock substrate, 12—salt pans, 13—dunes [34]. European red list (vascular plants/medicinal plants)—European red list of vascular plants [36]/European Red List of Medicinal Plants [37]. Polish red list (ang. Polish red list of pteridophytes and flowering plants) [29]/Polska Czerwona Księga Roślin (ang. Polish Red Data Book of Plants) [30]. Ecological indicator values according to Ellenberg—Ecological indicator values of vascular plants of Poland [35]: L light, M soil moisture, R soil acidity, where: L1: Deep shade, L2: Moderate shade, L3: Half shade, L4: Moderate light, L5: Full light; F1: Very dry, F2: Dry, F3: Fresh, F4: Moist, F5: Aquatic; R1: Highly acidic soils, R2: Acid, R3: Moderate acidic, R4: Neutral, R5: Alkaline. Ogrody botaniczne i arboreta—Index Plantarum of Polish Dendrological Collections [40], Index Plantarum of Outdoors Cultivated Herbaceous Plants in Poland [41], Collections of protected law and endangered plants and of species protected by the Bern Convention in the Polish Botanical Gardens [42]. NCPGR—National Center for Plant Gene Resources (https://bankgenow.edu.pl/) [91]. KFGB—Kostrzyca Forest Gene Bank (https://www.lbg.lasy.gov.pl/) [97] Seed banks—Silesian Botanical garden, "Botanical Garden in Warsaw Powsin", AMU Botanical Garden, The Botanic Garden of UMCS, Mountain Botanical Garden in Zakopane.

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
