# Peer review of "The Ex Situ Conservation and Potential Usage of Crop Wild Relatives in Poland on the Example of Grasses"

_agronomy, doi:10.3390/agronomy11010094_

Round 1

Reviewer 1 Report

Interesting descriptive study that results from the recently published checklist of CWR of Poland, providing further insights about the Poaceae CWR. There are some parts of the text where further information should be provided or whether the ideas need to be clearer, for example the crop or CWR categories referred in the methodology (see below my suggestions). There is some information that I would like to see added in the Results and Discussion section, namely regarding the use of Polish grass CWR in crop improvement and the ex situ conservation of studied CWR (also see below for more details). Another point I would like to make is the fact that it seems the section on “The potential of CWR usage among grasses” is given more importance than the section on “CWR in the genetic improvement of other species”; as the value of CWR lies in their (potential) use in crop improvement, the fact that it may have other (direct) uses is added value but not the main point. And finally, the English can be slightly improved and I have already provided some suggestions listed below.

ABSTRACT

Line 17: delete the word “separate”.

Lines 23–25: presumably the uses referred in the text are direct uses of the wild species and not as use in crop improvement?

Lines 26–27: do the 35% and the 80% refer to number of taxa or number of accessions?

INTRODUCTION

Line 32: After “Crop wild relatives” I suggest adding the acronym CWR between brackets, from that point one you can use the acronym.

Line 35: presumably “within the cultivation” refers to “within the cultivated forms of crops” so I would make this clearer.

Line 38: I suggest deleting the word “extensive” this is subjective and all is needed is knowledge on the genetics.

Line 39: why “very difficult”? Nowadays, with technological advances, there is a lot of information out there; the problem is that the information is not always gathered in the same place.  So I would delete that last bit of the text.

Line 41: the reference of the quote is missing and needs to be added.

Line 46: the reference [4] is not appropriate in this context.

Lines 51–52: those lists are commonly called checklists.

Line 53: …among others; you need to add references to each of the countries.

Lines 54–55: need to add reference to that statement.

Line 61: not clear what “national level conservation estimations and recommendations to the Polish CWRs checklist” mean…

Lines 66–67: The meaning of the sentence starting with “Recognizing a crop wild relative as a…” is not clear.

Line 69: transfer of what, needs to make it clearer.

Line 87: replace “2º” by “2nd”.

Lines 89-90: I would introduce a new paragraph starting with “This paper presents all CWR…”.

Line 95: in the context of this paper what is the difference between gene bank and seed bank?

MATERIALS AND METHODS

Line 107: replace “has to be considered” by “was considered”.

Lines 110–112: sentences needs to be re-written as it is not clear.

Line 112: replace “In the same study taxa” by “In the same study, taxa”.

Line 112: make clear what “1a category” means. Also, I do not understand why the wild relatives of popular grasses are not considered in this study.

Lines 114–119: it is not completely clear whether the authors refer to crop species or CWR in these lines. Do the resources referred in these lines include checklists of crops or of wild plant species from which CWR were extracted? Also, in the last sentence: if they are crop species, they might not occur in the wild, unless they have also wild populations. So this whole paragraph needs to be clearer.

Line 120: when “The status in Polish Flora” is referred, what does this mean exactly? The status in what?

Line 120: a quotation mark before “Flowering plants…2 is missing.

Lines 120–123: merge them into one single paragraph.

Line 125: unclear what “verified by category” means so the sentence needs to be re-written.

Line 129: the potential usage of the CWR refers to the direct use of the species so this needs to be clearer in the text.

Lines 130–132, 135, 139: dashes are different in the three lines.

Lines 143-151: the criterion you mention here, namely the presence in genebanks, in botanical gardens, is generally named “conservation status” or “ex situ conservation status”, whereas the last one (in the last lines of the paragraph) refers to legislation.

Line 152: replace “A taxon map…. were drawn” by “Taxa maps… were drawn”.

Lines 153-154: The sentence starting with “The maximum number (MAX)” probably needs to be re-written so it is clearer what “best square” means.

Lines 164: need to be more specific about the variables used, e.g. light but what exactly and which is the unit used.

Line 164: I suggest replacing “The hierarchical cluster analysis was applied…” by “A hierarchical cluster analysis was performed…”.

Lines 166-167: I suggest replacing “The multivariate statistics method, PCA…” by “A PCA (Principal… was carried out…”.

Line 168: variables are mentioned in this line, but which variables?

RESULTS AND DISCUSSION

I suggest comparing these results with those obtained in the paper by Jarvis S, Fielder H, Hopkins J, Maxted N and Smart S (2015) Distribution of crop wild relatives of conservation priority in the UK landscape. Biological Conservation 191: 444–451.

Appendices should be mentioned in the text, whenever appropriate, e.g. line 396 where it is mentioned that 80% of Polish CWR grasses are maintained in botanic gardens.

Section on “CWR in the genetic improvement of other species” should be moved to before the section on “The potential of CWR usage among grasses”. If the authors have not done so, I also suggest consulting the Harlan and de Wet inventory (https://www.cwrdiversity.org/checklist/) where the potential and/or confirmed uses are listed for some globally priority CWR. It is important to highlight the use of CWR in crop improvement because this is where the importance of CWR lies. The fact that they may be also directly used (for forage, food, medicinal purposes, etc) is a bit secondary. I would like to see more information about the use of these Polish CWR in plant improvement, whether they have been used nationally.

Section on “The potential of CWR usage among grasses”: I suggest changing the title to “The potential of grasses CWR direct use”: in the second sentence of this section, the authors refer to the actual or potential use in breeding but the following text mostly refers to the direct use of the species and not its use in crop improvement.

Section on “The ex situ conservation in gene banks and botanical gardens”: the idea that enough genetic diversity within each species should be conserved ex situ is not conveyed in this section and this is extremely important. A gene bank may have 1000 accessions of a particular population of a CWR and this may not be representative of the diversity found in the wild. Additionally, I would also like you to add whether studied grasses are conserved in international gene banks and for this purpose you could consult Genesys PGR (https://www.genesys-pgr.org/). Finally, I think it is important to raise the issue that botanic gardens have usually a few plants displayed which are not a genetic representation of the populations in the wild.

Line 173: I suggest deleting the word “taxa in this line.

Line 174-175: replace “extinct, and which” by “extinct, which”.

Line 183-184: replace “the genus Festuca” by “Festuca” only.

Lines 200-201: is it important to refer that the endemic CWR are considered separate species? It seems a bit redundant.

Line 247: there is something wrong here: “built "light value". (r= 0,79)”.

Line 325-326: replace “The aforementioned species are usually predominates and are…” by “The aforementioned species usually predominate and are…”.

Line 401: replace “preserve taxa” by “conserved taxa”.

CONCLUSIONS

Line 407: replace “grass’ species” by “grass species”.

Line 408: replace “CWR-flora” by “CWR flora”.

APPENDIX A:

  • A reference to the original publication where this checklist was extracted to should be added.
  • Information about the use of each CWR in crop improvement should be added.
  • “Potential use” should be changed to “Potential direct use”.

LEGENDS OF FIGURES

Figure 1: Maybe use “Occurrence status” rather than only “status”.

Reviewer 2 Report

Title is a bit clumsy.

Better would be – “The ex situ conservation and potential use of crop wild relatives in Poland using the example of grasses”

23-24 – “…huge potential for different usages in serving humans …”

Better to say ‘have the potential for many different uses in terms of the eco-system service benefits they can impart for humans, animals and the environment …’

24 – ‘biomass plants’ – perhaps better to say - ‘biomass fuel’.

25 – What is meant by ‘ecological’ in this context?

25-26 – ‘… crop wild relative germplasm is represented by 35% grasses …’

28 – ‘a large range of ex situ collections useful for the preservation, breeding and research of crop wild relatives, along with the necessary information …’

29 – What is meant by ‘tools’ in this context?  

30 – Surely ‘crop wild relatives’ ought to be a keyword.

32-33 – Does this refer to ‘landraces’ too? They could be consider to be ‘crop ancestors’, but perhaps not CWR.

35 – I’m not sure this makes sense – within cultivation, yes, but I’m not sure what is meant here ‘among crop wild relatives’? Maybe ‘… the importance of maintaining genetic diversity of crop plants in cultivation, and making use of the crop wild relatives to enhance this diversity through breeding’.

48  - ‘… adjust plants to climate changes…’ sounds too woolly. Better to say – ‘adapt crops to better withstand climate change…’

50 – ‘new crops’ – does this mean ‘new crop varieties’?

55 – ‘lower’ – better to say ‘reduce’.

55 - No need for the word ‘our’.

58 – What is meant by ‘engineering diversity’ – perhaps delete and just say ‘… gene editing, advantageous traits…’

84 – ‘…much more…’ replace with ‘many more’.

85 – ‘achieved’ – better just to say ‘had’.

87 – ‘2nd half’

93 – ‘appointed’ – Does this mean ‘named’?

112 – ‘belong’ – should be ‘belonging’.

123 – ‘Elleberg’ – should be ‘Ellenberg’.

140 – ‘or/and’ – better to be ‘and/or’.

150 – What does the abbreviation ang. mean?

160 – ‘draw’ – should be ‘drawn’.

172 – ‘belongs to’ – better to use simply ‘is one of the most…’

174 – Could say ‘locally extinct’ or ‘regionally extinct’

212- ‘at’ replace with ‘in the’

227 – Typo – ‘consists’

234 & 266 – Should be ‘Ellenberg’ or ‘Ellenberg’s’

269 – Simply ‘uses’

275 - ‘humans’

276 - ‘These’

279 – ‘fed’

280 – ‘where’

286 - ‘as an’

327 – ‘cricket’

407 – ‘grass’ – common overcomplication of non-native speaker.

Annex p14 – ‘potential’ and ‘preferred’

Some aspects are a little vague, particularly the final section on ex situ conservation. After mentioning this in the introduction, it seems to have been almost an after-thought, with much of the text

In the section on ex situ genebanks, I would have liked to have seen more recommendations of how to achieve this. Many of the statements in this section are obvious, and almost don't need to be mentioned. But, how are we going to achieve the goal of ensuring material is available in genebanks? Cross-border initiatives in continental Europe? Greater buy-in from FAO, IUCN etc.? More drive (and investment) from national and international governments? And, crucially, what are the consequences of not taking this action?

This paper adds some more recent perspective - Engels, J.M.M.; Thormann, I. Main Challenges and Actions Needed to Improve Conservation and Sustainable Use of Our Crop Wild Relatives. Plants 20209, 968.
